# Neuro-Spectral Architectures for Causal Physics-Informed Networks

**Arthur Bizzi**[1*], **Leonardo Moreira**[3], **Márcio Marques**[2], **Leonardo Mendonça**[2],
**Christian Oliveira**[2], **Vitor Balestro**[2], **Lucas Fernandez**[4], **Daniel Yukimura**[2],
**Pavel Petrov**[2], **João M. Pereira**[5], **Tiago Novello**[2], **Lucas Nissenbaum**[2]

[1]École Polytechnique Fédérale de Lausanne (EPFL), [2]Instituto de Matemática Pura e Aplicada (IMPA),
[3]Universidade do Estado do Rio de Janeiro (UERJ), [4]Laboratório Nacional de Computação Científica (LNCC),
[5]University of Georgia (UGA)

## Abstract

Physics-Informed Neural Networks (PINNs) have emerged as a powerful framework for solving partial differential equations (PDEs). However, standard MLP-based PINNs often fail to converge when dealing with complex initial value problems, leading to solutions that violate causality and suffer from a spectral bias towards low-frequency components. To address these issues, we introduce NeuSA (**Neu**ro-**S**pectral **A**rchitectures), a novel class of PINNs inspired by classical spectral methods, designed to solve linear and nonlinear PDEs with variable coefficients. NeuSA learns a projection of the underlying PDE onto a spectral basis, leading to a finite-dimensional representation of the dynamics which is then integrated with an adapted Neural ODE (NODE). This allows us to overcome spectral bias, by leveraging the high-frequency components enabled by the spectral representation; to enforce causality, by inheriting the causal structure of NODEs, and to start training near the target solution, by means of an initialization scheme based on classical methods. We validate NeuSA on canonical benchmarks for linear and nonlinear wave equations, demonstrating strong performance as compared to other architectures, with faster convergence, improved temporal consistency and superior predictive accuracy. Code and pretrained models are available in `https://github.com/arthur-bizzi/neusa`.

## 1 Introduction

The introduction of Physics-Informed Neural Networks (PINNs) [1] has sparked interest in using neural networks to solve partial differential equations (PDEs) [2, 3, 4]. PINNs enable data-efficient modeling of complex systems by embedding physical laws directly into the loss landscape. This approach has opened new possibilities in scientific computing, with applications spanning a wide range of subjects, including fluid dynamics and climate modeling [5, 6], biomedical simulations [7, 8], material science [9, 10, 11], and others [12, 13, 14, 15, 16]. We consider the following initial-boundary value problem for $t \in [0, T]$ and $\mathbf{x} \in \Omega \subset \mathbb{R}^d$:

$$\frac{d}{dt}\mathbf{u}(t, \mathbf{x}) = \mathbf{F}\left(t, \mathbf{x}, \mathbf{u}, \nabla\mathbf{u}, \nabla\nabla\mathbf{u}\right), \quad \mathbf{u}(0, \mathbf{x}) = \mathbf{u}_0(\mathbf{x}) \tag{1}$$

where $\mathbf{u} : [0, T] \times \Omega \to \mathbb{R}^n$ denotes the (vector-valued) solution, $\nabla\mathbf{u}$ and $\nabla\nabla\mathbf{u}$ denote its first- and second-order spatial derivatives, and $\mathbf{F}$ is a smooth function. $\mathbf{u}_0(\mathbf{x})$ is the initial condition (Cauchy data), and we assume some boundary conditions are imposed on $\partial\Omega$. PINNs consist of

---

*Corresponding author: arthur.coutinhobizzi@epfl.ch

39th Conference on Neural Information Processing Systems (NeurIPS 2025).

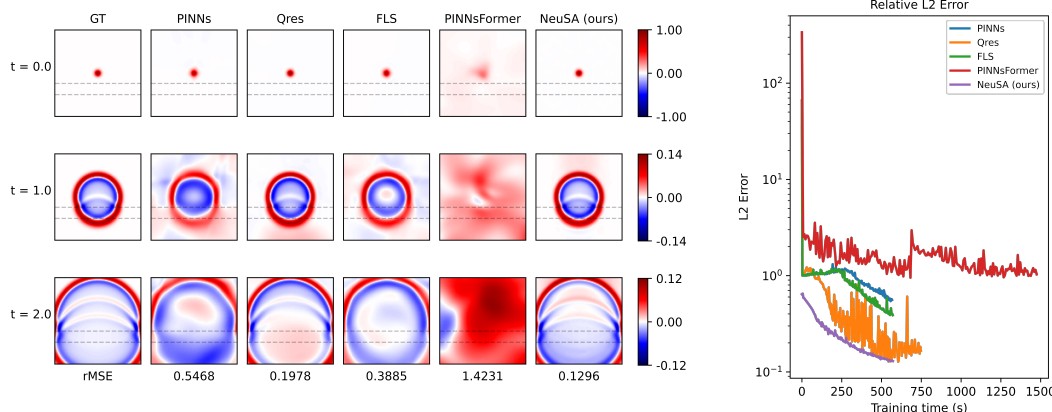

Figure 1: We present **NeuSA**, a theoretically grounded Physics-informed neural architecture. On the left, we compare various models on a wave propagation problem. The dashed lines represent the discontinuities of a stratified heterogeneous medium. NeuSA achieves the lowest relative error (rMSE) and most accurately preserves sharp wavefronts and reflections. On the right, we show the evolution of the relative L2 error during training. NeuSA converges more rapidly and consistently.

$\theta$-parametrized coordinate networks $\mathbf{u}_\theta : [0, T] \times \Omega \mapsto \mathbb{R}^n$, trained to approximate the solution $\mathbf{u}$ by minimizing a composite *physics-informed* loss function $\mathcal{L}(\theta)$. This loss typically includes terms for the PDE as well as for the initial/boundary data, e.g.,

$$\mathcal{L}_{\text{PDE}}(\theta) = \left\| \frac{d}{dt}\mathbf{u}_\theta(t, \mathbf{x}) - \mathbf{F}\left(t, \mathbf{x}, \mathbf{u}_\theta, \nabla\mathbf{u}_\theta, \nabla\nabla\mathbf{u}_\theta\right) \right\|_2^2 , \quad \mathcal{L}_{\text{IC}}(\theta) = \lambda_{\text{IC}} \left\| \mathbf{u}_\theta(0, \mathbf{x}) - \mathbf{u}_0(\mathbf{x}) \right\|_2^2 , \quad (2)$$

where $\| \cdot \|_2$ denotes the regular norm in $L^2([0, T] \times \Omega)$, and $\lambda_{\text{IC}}$ is the weight for the associated residue term (a term $\mathcal{L}_{\text{BC}}$ representing boundary conditions could be also included). PINNs thus provide a flexible, *data-driven*, and *meshless* framework for approximating PDE solutions using neural networks.

**Data-Driven**. As neural networks, PINNs are particularly suited for handling heterogeneous, noisy, or incomplete measurement data, which can be efficiently combined with physical priors via physics-informed losses [14, 15, 17, 18, 19].

**Mesh-Independence**. As coordinate networks, PINNs represent continuous interpolations of the underlying solutions and may be evaluated at arbitrary spatial or temporal coordinates. In higher-dimensional problems, this property can be combined with random sampling strategies to substantially reduce the number of samples needed to approximate the solution [20, 3].

However, standard PINNs often struggle to enforce fundamental structural aspects of the underlying solutions. In most cases, they rely on general-purpose feed-forward architectures, such as the standard Multi-Layer Perceptron (MLP) [21], or on specialized MLP-based variants that enhance expressivity through activation-function modifications, including QRes [22], FLS [23]. This generic structure design often leads to issues related to *spectral bias*, *causality* and limited *generalization capacity*:

**Spectral Bias**. Regular coordinate networks based on sigmoid or rectifier activations often struggle to represent high-frequency components, leading to issues with representing detailed and/or multi-scale solutions [24, 25]. This effect is often mitigated with Fourier-Feature (FF) layers, sinusoidal encoder layers designed to inject high-frequency representations into a network's architecture [26]. Still, FF layers require fine-tuning to avoid overfitting and noise.

**Causality**. PINNs are notorious for violating causality and temporal consistency due to their simultaneous training over the entire time domain [17]. These issues may manifest in the form of incorrect initial conditions or non-physical convergence to trivial solutions. Attempts have also been made to minimize these effects with modified losses [17, 19].

**Generalization Capacity**. MLP-based PINNs may struggle with extrapolation beyond their training domain [27, 28], which has been tackled with alternative training strategies [29].

Due to these issues, PINNs often fail to converge to the true solution when solving complex time-dependent problems. Instead, they may overfit and converge to trivial equilibrium solutions. Such shortcomings are common when solving problems with strong time dependence, as evidenced by their relative lack of success when applied to linear and nonlinear wave equations [12, 30, 31]. This stands in stark contrast to PINNs' capacity for solving parabolic and elliptic equations [32].

We propose **NeuSA** a new family of **Neu**ro-**S**pectral **A**rchitectures designed for solving space-inhomogeneous and/or nonlinear time-dependent PDEs. NeuSA uses the spectral decomposition to obtain a method-of-lines [33, 34] discretization of a PDE into a large system of ODEs, which is then modeled using a Neural ODE (NODE) [35] (see Figure 2). Figure 1 showcases NeuSA's results on a 2D wave propagation task, demonstrating significant improvements over prior methods in accuracy, speed, and temporal consistency. Our contributions may be summarized as follows.

- *Causality.* NeuSA is a spectral-method-based architecture for neural PDEs, such as [36]. Consequently, it inherits the causal structure of classical methods, including exact initial conditions and uniqueness, while retaining a data-friendly, mesh-less representation.

- *Spectral fidelity.* The choice of global spectral bases allows NeuSA to overcome the spectral bias commonly attributed to MLP-PINNs, offering a theoretically-motivated alternative to Fourier-Feature Networks.

- *Analytical initialization.* The interpretable structure of NeuSA as a neural extension of spectral methods enables specialized initialization schemes in which networks are initialized as the solution of closely related linear homogeneous problems, at no training cost.

- *Time-extrapolation.* Due to its causal formulation, NeuSA displays strong time-extrapolating performance, enabling simulation beyond training intervals.

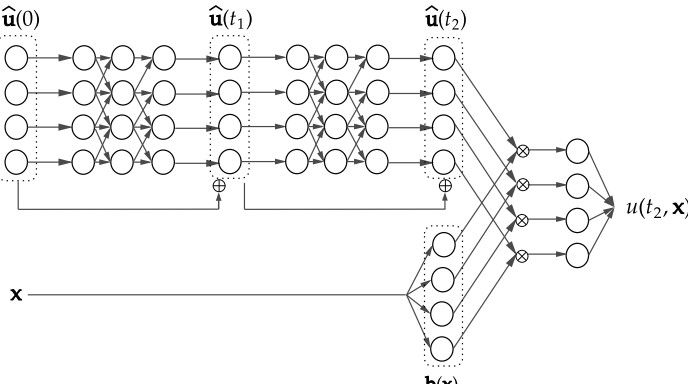

Figure 2: Neuro-Spectral Architecture. Above: The spectral coefficients for the initial conditions $\hat{\mathbf{u}}(0)$, flowing according to a NODE. Below: The spatial input $\mathbf{x}$ being encoded into the spectral basis functions $\mathbf{b}(\mathbf{x})$. Coefficients and bases are then combined to yield the final result.

## 1.1 Related work

Several works have explored designs for physics-informed neural machine learning, including a large body of literature on operator learning [37, 28]. Multiple recent works on Physics-Informed Networks propose alternative architectures for representing solutions, enhancing their expressiveness, spectral representation, or temporal coherence.

Quadratic Residual networks (**QRes**) [22] introduce a class of parameter-efficient neural networks by incorporating a quadratic term into the weighted sum of inputs before applying the activation functions at each layer of the network. This modification enables QRes to approximate polynomials using shallower networks, resulting in compact yet expressive models.

First-Layer Sine (**FLS**) [23], also referred to as sf-PINN, introduces sinusoidal encoding layers within the PINN framework. FLS nets utilize sinusoidal encoding layers, in an approach closely analogous to that of Fourier-feature networks, to mitigate spectral bias and enhance input gradient distribution.

**PINNsFormer** [4] adapts the transformer architecture to the PINN setting, leveraging attention mechanisms to model temporal dependencies among state tokens, with the goal of achieving enhanced temporal consistency.

Neural Ordinary Differential Equations (**NODEs**) [35] are a class of 'continuous-depth' residual networks that model inference as the integration of a continuous-time process, effectively solving an ODE whose dynamics are parameterized by a neural network. NODEs have proven powerful for modeling continuous-time dynamics and have been applied to physics-informed learning, generative modeling, time-series forecasting, and morphing [38, 39, 40, 41, 42, 43]. However, their highly sequential numerical structure makes them equivalent to ultra-deep residual networks, which can significantly slow down training. As a result, their application to PDEs remains relatively underexplored [44].

## 2 Neuro-spectral architectures

Neuro-Spectral architectures are defined as models that employ spectral decomposition to reduce a PDE defined over an infinite-dimensional space to an ODE system in finite dimensions, subsequently training a NODE to approximate the latter using a physics-informed loss. This formulation interprets (1) as an abstract Cauchy problem over the Hilbert space $L^2(\Omega)^n$, treating the solution $\mathbf{u}(t, \cdot)$ as a time-parametrized family $\mathbf{u} : \mathbb{R} \to L^2(\Omega)^n$. The spectral decomposition [45, 46, 47] consists of approximating $\mathbf{u}(\mathbf{x}, t)$ by its projection onto the subspace spanned by a finite subset $\mathbf{b}(\mathbf{x})$ of an orthonormal basis of $L^2(\Omega)^n$. Given a truncated spectral representation with harmonics $c_1, c_2, \ldots, c_d$, the solution $\mathbf{u}$ can be expressed in terms of an expansion over elements of the basis tensor $\mathbf{b}(\mathbf{x}) : \Omega \to \mathbb{C}^{c_1 \times \cdots \times c_d}$, whose coefficients form a tensor $\hat{\mathbf{u}}(t) : [0, T] \to \mathbb{C}^{c_1 \times \cdots \times c_d \times n}$, leading to

$$\mathbf{u}(t, \mathbf{x}) = \sum_k \hat{\mathbf{u}}_k(t) \mathbf{b}_k(\mathbf{x}), \quad \hat{\mathbf{u}}_k(t) = \int_\Omega \mathbf{u}(t, \mathbf{x}) \mathbf{b}_k(\mathbf{x}) d\mathbf{x}. \tag{3}$$

Where $k$ denotes a $d$-dimensional multi-index, $\hat{\mathbf{u}}_k(t) : [0, T] \to \mathbb{R}^n$ and $\mathbf{b}_k(\mathbf{x}) : \Omega \to \mathbb{R}$ represent the $k-$th indexed element in $\hat{\mathbf{u}}$ and $\mathbf{b}$, respectively. Substituting (3) into (1) leads to a method-of-lines [33, 34] discretization, resulting in an *ordinary* differential equation for the coefficients:

$$\frac{d}{dt}\hat{\mathbf{u}} = \hat{\mathbf{F}}(\hat{\mathbf{u}}), \tag{4}$$

where $\hat{\mathbf{F}} : \mathbb{C}^{c_1 \times \cdots \times c_d \times n} \to \mathbb{C}^{c_1 \times \cdots \times c_d \times n}$ usually does not admit a simple closed-form expression for a general basis $\mathbf{b}$. Instead of deriving it explicitly, we learn $\hat{\mathbf{F}}$ as a parameterized network $\hat{\mathbf{F}}_\theta$. Inference in a Neuro-Spectral model then proceeds as follows (see Figure 3 and [48]):

**1. Project the initial conditions onto an orthonormal basis.** Sample the initial conditions $\mathbf{u}(0, \mathbf{x})$ densely and extract their spectral representation $\hat{\mathbf{u}}$ in terms of the basis $\mathbf{b}$: $\mathbf{u}(0, \mathbf{x}) = \sum_k \hat{\mathbf{u}}_k(0) \mathbf{b}_k(\mathbf{x})$.

**2. Integrate coefficients in time according to a NODE.** Use the coefficients tensor $\hat{\mathbf{u}}$ as input to a NODE with vector field $\hat{\mathbf{F}}_\theta$ and integrate it with a high-order method: $\hat{\mathbf{u}}_\theta(t) = \hat{\mathbf{u}}(0) + \int_0^t \hat{\mathbf{F}}_\theta(\hat{\mathbf{u}}(\tau)) d\tau$.

**3. Reconstruct the solution and perform training.** Multiply the obtained coefficients $\hat{\mathbf{u}}_{\theta,k}(t)$ by their corresponding basis functions $\mathbf{b}_k(\mathbf{x})$ to obtain $\mathbf{u}(t, \mathbf{x})$. This representation can be differentiated analytically to compute physics-informed losses: $\mathbf{u}_\theta(t, \mathbf{x}) = \sum_k \hat{\mathbf{u}}_{\theta,k}(t) \mathbf{b}_k(\mathbf{x})$.

### 2.1 Spectral decomposition and initialization

The choice of the basis $\mathbf{b}$ can enforce specific properties of the solution, as well as ensure the fulfillment of the given boundary conditions. In this work, we initialize $\mathbf{b}$ as the Fourier basis and its odd and even extensions in terms of the sine and cosine functions. This choice enables the representation of homogeneous periodic, Dirichlet, and Neumann boundary conditions for rectangular domains. The spectral projection $\hat{\mathbf{u}}$ can then be computed in several ways, depending on how the spatial domain is sampled (see Appendix B). We adopt the Fourier basis for two main reasons: first, to overcome spectral bias, inspired by the success of Fourier-Feature layers; and second, to allow for the simple and accurate representation of linear translation-invariant (LTI) differential operators as scalar multipliers [49]. We use the latter to implement an improved initialization scheme for the NODE, described as follows.

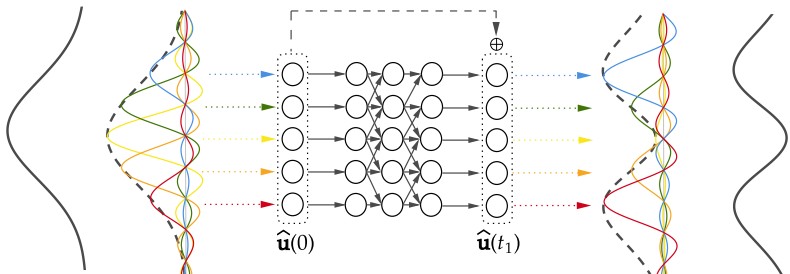

Figure 3: Inference in a Neuro-Spectral model. The initial conditions are decomposed into their spectral coefficients, which are propagated in time via a NODE. The time-iterated coefficients are then reconstructed into the solution at later times.

**1. Linearize the PDE.** Extract a linear translation-invariant approximation for $\mathbf{F}$:

$$\frac{d}{dt}\mathbf{u} \approx \mathbf{F}_{\text{linear}}(\mathbf{u}, \nabla\mathbf{u}, \nabla\nabla\mathbf{u}, \dots) := a_0\mathbf{u} + \sum_i a_{1i}\frac{d}{d\mathbf{x}_i}\mathbf{u} + \sum_{i,j} a_{2ij}\frac{d^2}{d\mathbf{x}_i d\mathbf{x}_j}\mathbf{u} + \cdots. \quad (5)$$

**2. Fourier multiplier.** Derive the associated Fourier multiplier $M \in \mathbb{C}^{c_1 \times c_2 \times \cdots \times c_d \times n}$, defined as an element-wise polynomial on the $k$-th Fourier frequency corresponding to the $k$-th harmonic:

$$\frac{d}{dt}\mathbf{u} \approx \mathbf{F}_{\text{linear}} \implies \frac{d}{dt}\hat{\mathbf{u}} \approx M \odot \hat{\mathbf{u}}, \quad (6)$$

where $\odot$ stands for the Hadamard (element-wise) product.

**3. Initialize the vector field $\hat{\mathbf{F}}_\theta$.** Initialize NODE near this approximate linear solution by augmenting the learned vector field with the analytical multiplier:

$$\hat{\mathbf{F}}_\theta(\hat{\mathbf{u}}) = (M \odot \hat{\mathbf{u}}) + \epsilon\mathcal{F}_\theta(\hat{\mathbf{u}}), \quad (7)$$

where $\mathcal{F}_\theta(\hat{\mathbf{u}})$ is a neural network initialized with mean zero and unit variance, and $\epsilon$ is a small parameter. In this way, the network starts close to the solution of the associated LTI problem, serving as a strong prior for training. During optimization, $\mathcal{F}_\theta$ learns a compact representation for the non-linear and/or non-translation-invariant dynamics, effectively leading to a neural generalization of the classical spectral method. See Section 3 for explicit examples, and Appendix B.2 for the detailed architecture of $\mathcal{F}\theta$.

## 2.2 Neural ODE, time integration and causality

As discussed, neuro-spectral models rely on a NODE to propagate the spectral coefficients forward in time. The vector field to be integrated over as part of inference is composed of the near-analytical initialization discussed above along with a multilayer perceptron $\mathcal{F}_\theta$:

$$\frac{d}{dt}\hat{\mathbf{u}} = \hat{\mathbf{F}}_\theta(\hat{\mathbf{u}}) = M \odot \hat{\mathbf{u}} + \epsilon\mathcal{F}_\theta(\hat{\mathbf{u}}). \quad (8)$$

The NODE receives as input a tensor of size $1 \times c_1 \times c_2 \times \cdots \times c_d \times n$, corresponding to the first time-slice of the solution, and outputs $t_{\text{samples}}$ slices in the form of a tensor of shape $t_{\text{samples}} \times c_1 \times c_2 \times \cdots \times c_d \times n$. This special treatment of the time dimension, characterized by the sequential nature of integration, is what equips NeuSA with causal structure. In fact, it is possible to prove NeuSAs are *flows* [50, 51], as summarized in the following theorem:

**Theorem 1.** *For band-limited initial conditions $\mathbf{u}_0$ and globally Lipschitz neural vector fields $\hat{\mathbf{F}}_\theta$, the orbits created by NeuSA satisfy the initial conditions and uniqueness:*

    *1. fulfillment of initial conditions: $\mathbf{u}_\theta(0, \mathbf{x}) = \mathbf{u}(0, \mathbf{x})$;*

    *2. uniqueness: $\mathbf{u}_\theta^1(0, \cdot) \neq \mathbf{u}_\theta^2(0, \cdot) \implies \mathbf{u}_\theta^1(t, \cdot) \neq \mathbf{u}_\theta^2(t, \cdot) \quad \forall t \in [0, T]$.*

A more detailed exposition of this theorem as well as its proof may be found in Appendix A. It is, in essence, a result of the properties of flow operators for ODEs combined with the uniqueness of the spectral decomposition for band-limited functions. This holds *by construction*, regardless of training.

## 2.3 Losses and training

Training NeuSA is similar to training a common MLP-PINN. The main difference is that we can no longer differentiate directly with respect to time, as it is no longer an input coordinate; it is instead implicitly encoded as the time-steps for the NODE iteration. Nevertheless, time and space derivatives may be calculated in a straightforward manner:

$$\frac{d}{dt}\mathbf{u}_\theta(t,\mathbf{x}) = \sum_k \hat{\mathbf{F}}_\theta(\hat{\mathbf{u}}_\theta)_k(t)\mathbf{b}_k(\mathbf{x}), \quad \frac{d}{d\mathbf{x}_i}\mathbf{u}_\theta(t,\mathbf{x}) = \sum_k \hat{\mathbf{u}}_{\theta,k}(t)\frac{d}{d\mathbf{x}_i}\mathbf{b}_k(\mathbf{x}), \tag{9}$$

where by construction $\hat{\mathbf{F}}_\theta(\hat{\mathbf{u}}_\theta)_k(t) = \frac{d}{dt}\hat{\mathbf{u}}_{\theta,k}(t)$. Note that the cost of calculating derivatives does not increase meaningfully for higher-order spatial derivatives, as opposed to the exponential increase in computational cost incurred by naively stacking derivatives with autograd [52]. We may then sample the domain $\Omega$ and evaluate the associated Physics-Informed residue with

$$\mathcal{L}_{\text{PDE}}(\theta) = \sum_{t_i\in[0,T]} \sum_{\mathbf{x}_j\in\Omega} \left\| \frac{d}{dt}\mathbf{u}_\theta(t_i,\mathbf{x}_j) - \mathbf{F}(t_i,\mathbf{x}_j,\mathbf{u}_\theta,\nabla\mathbf{u}_\theta,\nabla\nabla\mathbf{u}_\theta) \right\|_2^2, \tag{10}$$

where $t_i$ denotes the $i$-th integration time step for the NODE and $\mathbf{x}_j$ denotes the coordinates at the $j$-th spatial sample point. Note that NeuSA automatically complies with initial and boundary conditions and therefore does not require loss terms for them.

Note that neither time nor space samples need to be uniformly distributed; nevertheless, space samples must remain constant across all times for each pass, as opposed to conventional PINNs. This comes at the advantage that a *single* forward pass is necessary to evaluate the loss over all samples, as opposed to the multiple passes needed for common PINNs. This will allow NeuSA architectures to achieve training speeds comparable to those of purely neural approaches, despite their reliance on computationally intensive NODE integration.

## 3 Experiments

We evaluate NeuSA on boundary and initial value problems for three PDEs: the 2D wave equation, the 2D Burgers' equation, and the 1D nonlinear sine–Gordon equation. In all cases, we address the forward (direct) problem, where the models are trained to learn approximate solutions given known conditions. We compare the performance and accuracy of NeuSA against several established MLP-based PINN architectures: the original PINN [1], QRes [22], FLS [23], and PINNsFormer [4].

**Training setup.** NeuSA and all baseline models are implemented in PyTorch [53]. The baseline configurations follow the setups described in PINNsFormer [4] and RoPINN [3]. PINN, QRes, and FLS are initialized using Xavier initialization [54], with the hyperbolic tangent as the activation function (except for the first FLS layer, which acts as a Fourier feature mapping [55, 26]). All remaining hyperparameters were tuned to achieve the best performance for each model (e.g. weights for the initial and boundary condition losses). For the NODEs used in NeuSA, we adopt the implementation given by the TorchDyn library [56]. The vector fields $\mathcal{F}_\theta$ are modeled as MLPs with dimensionwise layers (see Appendix B.2) with two hidden layers, ReLU activations, and Glorot initialization, and are integrated using a fourth-order Runge–Kutta solver. Gradients are computed via standard backpropagation through the ODE solver.

Training is performed using the Adam optimizer [57]. NeuSA's strong architectural priors enable the use of larger learning rates compared to the baseline models, which are trained with the recommended rate of $10^{-3}$ (see §6.1 of [58]). All baseline models exhibited reduced performance when trained with learning rate $10^{-2}$, due to instability. Each experiment was run several times, and the mean of each evaluation metric was reported. To evaluate model accuracy, we consider the standard rMSE (*relative mean squared error*, also called the *relative $L_2$ error*) and rMAE (*relative mean absolute error*, also called the *relative $L_1$ error*) metrics. The ground-truth solution for each problem is obtained with high accuracy numerical solvers (see Appendix C). Experiments were executed on an Nvidia RTX 4090 GPU (24 GB VRAM). Results are summarized in Table 1.

Additional details and extended results for each experiment are provided in Appendix C, and additional experiments exploring NeuSA's training and inference time may be found in Appendix D.

## 3.1 Wave equation: 2D Layers, 3D Layers and Marmousi

As our first benchmark, we consider an initial-value problem for the linear wave equation in an infinite heterogeneous medium, a canonical problem in acoustics, seismics, and electromagnetism, [59, 60, 61]:

$$\frac{\partial^2}{\partial t^2}\mathbf{u} = c^2(\mathbf{x})\Delta\mathbf{u}, \ \ \mathbf{u}(\mathbf{x},0) = \exp\left(-\frac{|\mathbf{x}|^2}{2\sigma^2}\right), \ \ \frac{\partial}{\partial t}\mathbf{u}(\mathbf{x},0) = 0,\tag{11}$$

where $\Delta$ denotes the Laplacian in the spatial dimensions, $\sigma = 0.1$, and $c$ denotes the material-dependent wave speed. This material heterogeneity leads to wave reflection and refraction at the interfaces between layers. We evaluate NeuSA in three scenarios of increasing complexity: the 2D wave equation for an environment with three horizontal layers (hereafter referred to as 2D Layers), the 3D wave equation in a medium with two horizontal layers (here referred to as 3D Layers), and the 2D Marmousi model. In all cases, the spatial domain is truncated to $[-2, 2]^d$ for $d = 2, 3$. This is a valid approximation to the infinite domain scenario as long as the propagating waves do not reach the domain boundaries.

For the 2D Layers and 3D Layers cases, the propagation medium consists of three and two horizontal layers, respectively. Such simplified test problems are frequently found in seismic datasets [62]. The Marmousi reservoir model [63] is a canonical benchmark with highly complex stratified medium in two dimensions, containing folds of rocks and an overlying water layer [64]. Images depicting both 2D media may be found in Appendix C.

For NeuSA, we initialize the neural vector field near the solution of the homogeneous wave equation and constrain $\mathcal{F}_\theta$ to take a low-rank form, as detailed in Appendix B. The model is trained for 2,000 steps, using $201^d$ basis elements and integrated over 201 time steps. The baseline models are trained for 20,000 steps, using 10,000 random collocation points for the PDE and 1,000 points for the initial conditions. See Figure 1 for an image depicting the results obtained for the 2D Layers and Figure 4 for the Marmousi reservoir. The results obtained for the 3D Layers case are presented in Appendix C.

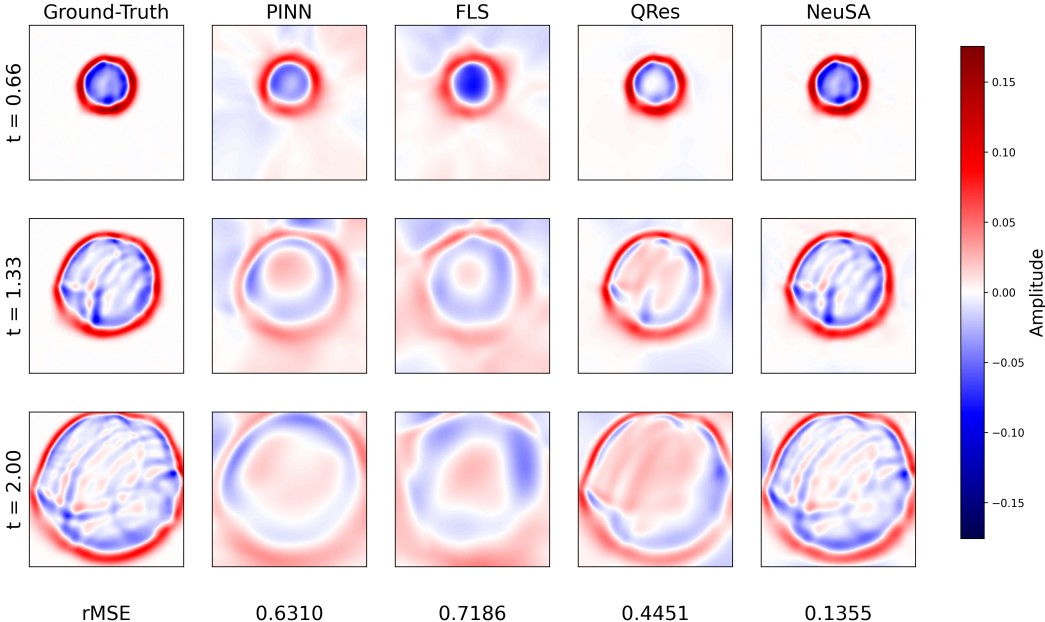

Figure 4: Results for the wave equation over the Marmousi benchmark. NeuSA is able to achieve a solution that is much closer than reference methods to the ground truth.

## 3.2 Sine-Gordon equation

In this example, we consider the 1D sine-Gordon (s-G) equation, a generalized nonlinear wave equation that has applications in soliton collisions and inverse scattering [65]. Consider the problem

$$\frac{\partial^2 \mathbf{u}}{\partial t^2} = \frac{\partial^2 \mathbf{u}}{\partial x^2} - 10\sin(\mathbf{u}), \quad \mathbf{u}(0,\mathbf{x}) = \frac{1}{\sqrt{2\pi}\sigma}\exp\left(-\frac{|\mathbf{x}|^2}{2\sigma^2}\right), \quad \frac{\partial}{\partial t}\mathbf{u}(\mathbf{x},0) = 0, \qquad (12)$$

with $(\mathbf{x},t) \in [-4,4] \times [0,3]$, zero-Dirichlet boundary conditions at $x = \pm 4$, and $\sigma = 0.1$.

For the numerical experiments, NeuSA is trained with 201 frequencies and 201 time steps, using $1,000$ steps with a learning rate of $0.01$. The PINN, QRes, and FLS models are trained on a regular grid of dimension $201 \times 201$, while a $101 \times 101$ grid is considered for the PINNsFormer model. All models other than NeuSA were trained for $10,000$ Adam steps. The computational results are visualized in Figure 5.

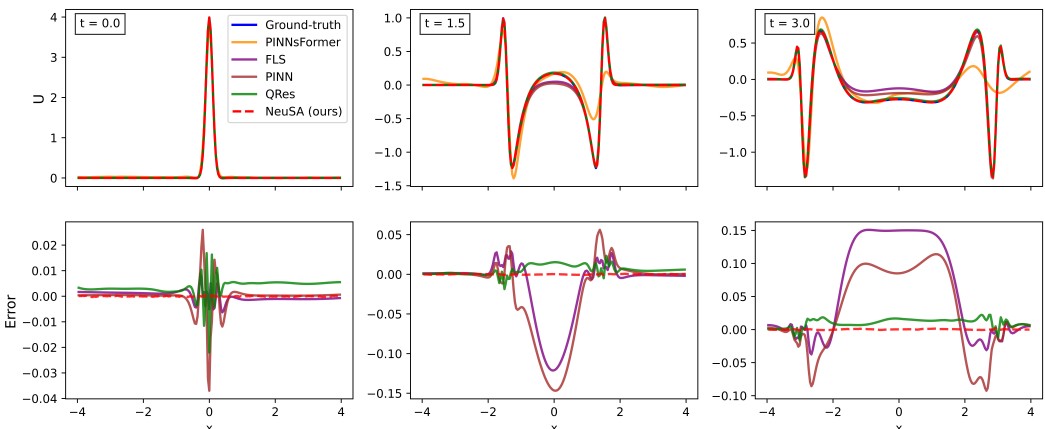

Figure 5: Results from the sine-Gordon equation. Above: the solution versus the ground truth. Below: the residues between them. NeuSA is both faster and more accurate than the baselines.

## 3.3 2D Burgers' equation

We address a 2D generalization of Burgers' equation, commonly used as a benchmark in computational fluid dynamics [66] to model the development of two-dimensional shocks [67]. We consider the initial-boundary value problem

$$\frac{\partial}{\partial t}\mathbf{u} = -\mathbf{u} \cdot \nabla\mathbf{u} + \nu\Delta\mathbf{u}, \quad \mathbf{u}(\mathbf{x},0) = \big(\sin(\pi x_1)\sin(\pi x_2), \cos(\pi x_1)\cos(\pi x_2)\big). \qquad (13)$$

where $\mathbf{u} = \big(u(\mathbf{x},t), v(\mathbf{x},t)\big)$, $(\mathbf{x},t) \in [0,4]^2 \times [0,1]$, and $\nu = 0.01$, and periodic boundary conditions in $\mathbf{x}$ are assumed.

NeuSA is trained for 200 steps using $201 \times 201$ components over 201 time steps, with a vector field initialized near the solution of the corresponding heat equation. The baseline models are trained for 20,000 steps, using 10,000 collocation points for the PDE, 1,000 for the initial condition, and 500 for the boundary condition.

We conducted an additional experiment on Burgers' equation to assess our method's ability to extrapolate the approximated solution in time beyond its training interval. All models were trained on the interval $[0,1]$ and then evaluated on the extended interval $[0,2]$. We analyzed performance in successive time instants, quantifying how prediction accuracy degrades as the time gets further from the training region. Because our method integrates a learned vector field over time, we expected it to extrapolate smoothly into future time, rather than exhibit the localized overfitting often observed in standard MLP-based architectures. The results shown in Figure 6 confirm our expectations.

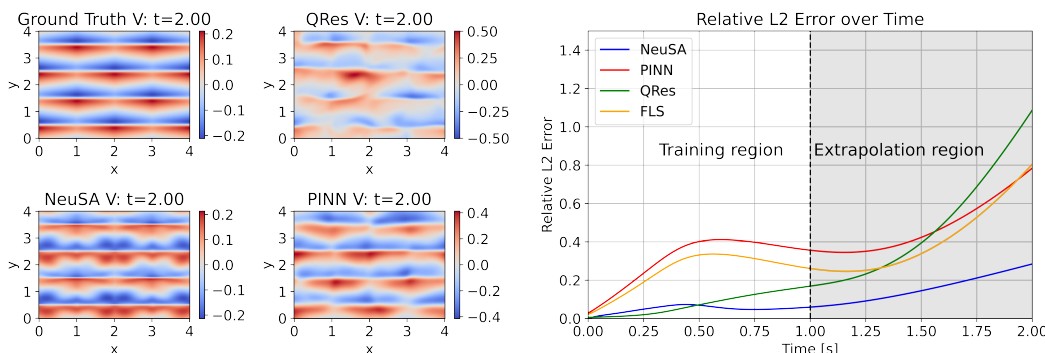

Figure 6: Extrapolation results for the Burgers' equation. Left: solutions from PINN and QRes degenerate strongly when extrapolating; in contrast, NeuSA's solution retains qualitative behavior observed in the Ground Truth. Right: NeuSA outperforms all the baselines when extrapolating, maintaining comparable performance. Other well performing models, such as QRes, quickly diverge.

Table 1: Average experiment metrics for each model. TT refers to the training time in seconds. Despite being generally trained for less time, NeuSA consistently matches or outperforms the baselines.

| Model | 2D Layers | | Marmousi | | 3D Layers | | 2D Burgers | | Sine Gordon | |
|---|---|---|---|---|---|---|---|---|---|---|
| | rMSE | TT | rMSE | TT | rMSE | TT | rMSE | TT | rMSE | TT |
| PINN | 0.545 | 566 | 0.698 | 635 | 0.073 | 5990 | 0.221 | 871 | 0.139 | 976 |
| QRes | 0.115 | 750 | 0.412 | 718 | 0.021 | 8775 | 0.073 | 1,135 | 0.020 | 1,315 |
| FLS | 0.590 | 577 | 0.684 | 648 | 0.070 | 6179 | 0.202 | 885 | 0.135 | 1,015 |
| PINNsFormer | 1.072 | 1484 | — | — | — | — | 1.053 | 2,294 | 0.681 | 3,333 |
| **NeuSA** | **0.075** | **530** | **0.171** | **573** | **0.008** | **702** | **0.051** | **112** | **0.001** | **215** |

## 4  Results and analysis

The results for the wave equation show that NeuSA can accurately propagate waves in complex media, leading to an error between one and two orders of magnitude smaller than the baselines. This is despite NeuSA being trained for 10 times fewer training steps. It is worth noting that a key challenge in seismic and acoustic simulations with heterogeneous media is to capture the complex wave patterns generated by reflections at material discontinuities. NeuSA is the only approach that accurately recovers second-order reflections, as shown in Figure 1, at $t = 2s$.

The results for the sine-Gordon equation in Figure 5 indicate that NeuSA very quickly learns the dynamics for the system with outstanding precision, obtaining error metrics an order of magnitude smaller than QRes, the next best result, with a substantially smaller training time. NeuSA's causal structure allows it to propagate the solution from initial conditions, staying in close agreement with the ground-truth. In contrast, PINN, FLS and PINNsformer fail to do so, despite being trained for 10 times more epochs.

The results for the Burgers' equation show that NeuSA successfully captures the qualitative behavior of the solution, including the formation of detailed two-dimensional shocks. Quantitative results in Table 1 show that NeuSA exhibits strong performance, while requiring substantially less training time, leading to gains over PINNs, FLS, PINNsFormer and QRes.

These results are underscored by the very strong performance in the extrapolation task shown in Figure 6, which demonstrates that NeuSA has learned a robust internal representation for the system's autonomous dynamics. As a result, NeuSA can estimate and extrapolate solutions over large times intervals beyond the training domain, leading to substantially superior performance as compared to the baselines. We attribute this capability to NeuSA's causal structure.

Although built on the inherently slower NODE framework, NeuSA trains significantly faster than MLP-based PINNs. This efficiency is likely due to the physical and causal priors built into the architecture, allowing for convergence in considerably fewer steps. Moreover, neuro-spectral models

compute the solution across the entire domain in a single forward pass, in contrast to the thousands of passes needed for a PINN, reducing the relative computational overhead. Appendix D has a longer discussion on this effect. Additionally, NeuSA produces remarkably accurate solutions, which we attribute to its spectral representation capabilities and its automatic compliance with initial and boundary conditions. NeuSA does not need to train on boundary data, allowing us to perform optimization only on the physics-informed loss. This helps avoid the conflicting gradients of data and equation losses, which often lead to unstable training, as shown in the rMSE-versus-time plots of the tested classical architectures in Figure 1.

Nevertheless, the method does have limitations, which we discuss in greater detail in Appendix E. First, all experiments take place on simple spatial domains, which allows using the framework of the Fourier bases; more complex geometries would require the implementation of more generic bases, which could hinder the initialization process. Second, NeuSA integrates the vector-valued solution $\hat{\mathbf{u}}(\mathbf{x}, t)$ using Runge-Kutta methods which, despite their efficiency, may become unstable when applied to stiff problems. Finally, NeuSA's performance seems to be strongly influenced by its analytical initialization procedure; while this approach enables exceptionally fast training when a suitable linear approximation is available, its effectiveness significantly diminishes when initialized with no prior dynamical information.

## 5  Conclusion

We have introduced NeuSA (**Neu**ro-**S**pectral **A**rchitectures), a novel class of PINNs grounded in numerical spectral methods and designed to solve time-dependent nonlinear PDEs in inhomogeneous media. These architectures overcome the spectral bias and causality issues associated with MLP-PINNs by construction. Furthermore, we have shown that the architecture inherently satisfies boundary and initial conditions, as well as ensures solution uniqueness, without incurring any additional training cost.

Based on the numerical experiments for the 2D wave equation in heterogeneous media, 2D Burgers' equation, and 1D nonlinear sine-Gordon equation, we observed that NeuSA can solve complex problems, often achieving smaller error with significantly shorter training time than reference physics-informed architectures. Such capabilities are likely due to the improved initialization procedure and causal structure of NeuSA, which results in more physically relevant solutions.

The development of NeuSA paves the way for future research and applications. In particular, the architectures can be adapted to solve other challenging PDEs, such as the widely studied Navier-Stokes equations [68], or even pseudodifferential equations arising in wave propagation [69, 70]. Although initializing with a Fourier basis has proven highly effective for the applications considered in this work, other bases could also be explored to enhance performance and, thus, make NeuSA less dependent on the mesh. Finally, other numerical methods could be used to replace the fourth-order Runge-Kutta and improve the overall performance of NeuSA, especially when applied to stiff problems.

### Acknowledgments

This work was supported by Petrobras. João Pereira is thankful for a start-up grant from the University of Georgia. We also acknowledge financial support from Google. We would also like to thank Deborah Oliveira and Lucas Schwengber for fruitful discussions.

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

## A  Causality of NeuSA

NeuSA is a "causal" architecture in the sense that it produces solutions which are *evolutionary* by nature, implying uniqueness and continuous dependence on initial conditions. It is generally impossible to prove the convergence of a numerical method to the solution of a general PDE without imposing strong restrictions upon the spectrum of initial conditions and the function $\mathbf{F}$. We may nevertheless prove that the solutions generated by our method have the properties associated with solutions to evolution problems. For the finite-dimensional system in eq. (4), these properties are encapsulated in the properties of the associated *flow operator* $\Phi : [0,T] \times \mathbb{C}^{c_1 \times \cdots \times c_d \times n} \to \mathbb{C}^{c_1 \times \cdots \times c_d \times n}$, defined as:

$$\Phi^t \hat{\mathbf{u}}_0 = \hat{\mathbf{u}}_0 + \int_0^t \hat{\mathbf{F}}(\hat{\mathbf{u}}(\tau))d\tau \,. \tag{14}$$

These operators have the following semigroup properties:

1. There exists an identity element: $\Phi^0 \hat{\mathbf{u}}_0 = \hat{\mathbf{u}}_0$ .
2. The flow is an additive group action: $\Phi^{t_2} \Phi^{t_1} \hat{\mathbf{u}}_0 = \Phi^{t_2+t_1} \hat{\mathbf{u}}_0$ .

Most importantly, these two properties translate immediately to the causal properties that we aim to prove: Property 1 implies that initial conditions are enforced, while Property 2 is equivalent to uniqueness [51]. Theorem 1 then follows as a consequence of the fact that NeuSA solutions define a flow.

Let $S_{\mathbf{b}} \subset L^2(\Omega)$ be the finite-dimensional subspace spanned by our basis elements $\mathbf{b}$. Consider now the decomposition operator $P : S_{\mathbf{b}} \to \mathbb{C}^{c_1 \times \cdots \times c_d \times n}$, an isomorphism mapping each element of $S_{\mathbf{b}}$ into the tensor of coefficients of its expansion over the elements of $\mathbf{b}$. We also define the reconstruction operator $P^\dagger : \mathbb{C}^{c_1 \times \cdots \times c_d \times n} \to S_{\mathbf{b}}$ mapping the coefficient tensor onto the respective linear combination of the vectors $\mathbf{b}_k$. Inference for NeuSA consists of the following steps:

1. obtaining expansion coefficients $\hat{\mathbf{u}}_0 = P\mathbf{u}_0$ of the initial data in the basis $\mathbf{b}$ (possibly, projecting $\mathbf{u}_0$ onto $S_{\mathbf{b}}$);
2. acting on $\mathbf{u}_0$ by the flow of the vector field $\mathbf{F}_\theta$ using NODE;
3. reconstructing the continuous solution via $\mathbf{u}(t) = P^\dagger \hat{\mathbf{u}}(t)$.

These steps can be summarized as

$$\mathbf{u}_\theta(t) = P^\dagger \Phi_\theta^t P\mathbf{u}_0 \,, \quad \text{where} \quad \Phi_\theta^t \hat{\mathbf{u}}_0 = \hat{\mathbf{u}}_0 + \int_0^t \hat{\mathbf{F}}_\theta(\hat{\mathbf{u}}(\tau))d\tau \,. \tag{15}$$

Now we can formulate and prove the following

**Theorem 2.** *For band-limited initial conditions $u_0 \in S_{\mathbf{b}}$ and globally-Lipschitz neural vector fields $\hat{\mathbf{F}}_\theta$, the orbits created by NeuSA satisfy the initial conditions and are unique:*

1. *fulfillment of initial conditions:* $\mathbf{u}_\theta(0, \mathbf{x}) = \mathbf{u}(0, \mathbf{x})$ ;

2. *uniqueness:* $\mathbf{u}_\theta^1(0, \cdot) \neq \mathbf{u}_\theta^2(0, \cdot) \implies \mathbf{u}_\theta^1(t, \cdot) \neq \mathbf{u}_\theta^2(t, \cdot) \quad \forall t \in [0, T]$ .

*Proof.* For band-limited functions $\mathbf{u} \in S_{\mathbf{b}}$, the decomposition $P$ and reconstruction $P^\dagger$ are bijections, and $P^\dagger$ is the inverse of $P$:

$$P^\dagger P\mathbf{u} = \mathbf{u} . \tag{16}$$

This fact immediately follows from uniqueness of expansion coefficients of any function $\mathbf{u} \in S_{\mathbf{b}}$ for the given basis $\mathbf{b}$. Likewise, for a globally Lipschitz neural network $\hat{\mathbf{F}}_\theta$, there exists a flow $\Phi_\theta$ associated with the NODE $d\hat{\mathbf{u}}/dt = \hat{\mathbf{F}}_\theta(\hat{\mathbf{u}})$ (i.e., the flow determined by the vector field $\hat{\mathbf{F}}_\theta(\hat{\mathbf{u}})$). Moreover, its orbits are unique. This follows from the classical existence and uniqueness of solutions for ordinary differential equations in Banach spaces [71, 51].

Property 1 then follows from the uniqueness of the spectral decomposition combined with the flow properties of $\Phi_\theta$:

$$\mathbf{u}_\theta(0) = P^\dagger \Phi_\theta^0 P\mathbf{u}_0 = P^\dagger P\mathbf{u}_0 = \mathbf{u}_0 . \tag{17}$$

Likewise, to prove property 2 we use the uniqueness of the spectral decomposition as well as the fact that NeuSA encodes the flow $\Phi_\theta$ (under which the orbits do not intersect), represented as follows:

$$P^\dagger \Phi_\theta^{t_2} P\mathbf{u}_\theta(t_1) = P^\dagger \Phi_\theta^{t_2} PP^\dagger \Phi_\theta^{t_1} P\mathbf{u}_0 = P^\dagger \Phi_\theta^{t_2+t_1} P\mathbf{u}_0 = \mathbf{u}_\theta(t_2 + t_1) . \tag{18}$$

$\square$

As mentioned above, in practice decomposition over the spectral basis $\mathbf{b}$ is preceded by the projection of the initial data onto $S_{\mathbf{b}}$. The basis $\mathbf{b}$ can always be chosen to be large enough to approximate any smooth function with the required accuracy. Likewise, it is generally reasonable to assume that common (finite-sized) feedforward networks are globally Lipschitz-continuous, since they are essentially compositions of Lipschitz maps and Lipschitz nonlinear activations.

## B   Implementation details

In this section, we provide further details concerning our code implementation. The construction of the Fourier basis together with the extraction of its coefficients is explained in Subsection B.1. Lastly, Subsection B.2 discusses the numerical scheme adopted to enhance dimension-wise layers for multiple space dimensions, since they enable efficient modeling of complex interactions across spatial dimensions while avoiding the prohibitive costs of fully dense layers.

### B.1   Spectral decomposition

To operate in the spectral domain, we represent the target functions using a spectral decomposition. The following subsections describe how the Fourier basis is constructed and how the corresponding coefficients are extracted.

### B.1.1   Constructing the basis

We assume for this section that the domain $\Omega$ can be decomposed as a direct product (e.g., into a direct product of 1D intervals), leading to the following product representation for the basis:

$$\mathbf{b}_k(\mathbf{x}) = \prod_{i=1}^d \mathbf{b}_{k_i}(\mathbf{x}_i) . \tag{19}$$

For example, the standard Fourier basis functions may be expressed as the products of the following form (up to a normalization constant):

$$\mathbf{b}_k(\mathbf{x}) \propto \prod_{i=1}^d \exp(i\omega_{k_i}\mathbf{x}_i) , \tag{20}$$

or, likewise, in terms of the respective sine/cosine functions, depending on the boundary conditions that have to be imposed.

In addition to enabling our initialization procedure, initializing $\mathbf{b}$ as the Fourier basis tends to result in dense representations for nonlinear and/or non translation-invariant dynamics [2] [72]. While in the context of numerical solution of PDEs this is often undesirable, in the context of neural networks this will become an advantage, as denser connections between coefficients should lead to (positive) overparametrization [73].

Nevertheless, the Fourier basis has limitations. Foremost among them is its strictly non-local nature, which often results in difficulties when reproducing highly localized PDE solutions.

### B.1.2 Extracting the coefficients û and evaluating derivatives

In order to perform the spectral decomposition $\mathbf{u} \mapsto \hat{\mathbf{u}}$, we sample the function over $N$ collocation points $\{\mathbf{x^i}\}_{i=0}^{N-1}$ and represent it in terms of $N$ basis terms. This may be expressed as:

$$\mathbf{u}(t, \mathbf{x}^i) = \sum_k \hat{\mathbf{u}}_k(t)\mathbf{b}_k(\mathbf{x}^i), \quad i \in 1, \ldots, N, \tag{21}$$

where $k$ is a $d$-dimensional multi-index with $N$ elements. This is a linear system that can be solved in a straightforward manner; for example, given a grid structure, the Discrete Fourier Transform and its variations may be used. As it is usually done in the literature on spectral methods, we can simplify the basis representation by reshaping the domain via a transformation $\chi$, leading to:

$$\mathbf{u}(t, \mathbf{x}^i) = \sum_k \hat{\mathbf{u}}_k(t)\mathbf{b}_k(\chi(\mathbf{x}^i)). \tag{22}$$

Differentiation then takes place as described in Section 2:

$$\frac{d}{d\mathbf{x}_j}\mathbf{u}(t, \mathbf{x}^i) = \sum_k \hat{\mathbf{u}}_k(t)\frac{d}{d\mathbf{x}_j}\mathbf{b}_k(\chi(\mathbf{x}^i)), \tag{23}$$

where the chain rule is used to calculate the right-most term. For a concrete example, a 2D problem expressed in the Fourier basis over a square domain $\Omega = [-L, L] \times [-L, L]$ may be mapped into the canonical domain $[-\pi, \pi] \times [-\pi, \pi]$ with the transformation $\chi(\mathbf{x}) = \pi\mathbf{x}/L$, leading to the following form for the derivatives:

$$\frac{d}{d\mathbf{x}_j}\mathbf{u}(t, \mathbf{x}^i) = \sum_k \frac{i\pi\omega_{k_j}}{L}\hat{\mathbf{u}}_k(t)\prod_{i=1}^d \exp(i\omega_{k_i}\mathbf{x}_i), \tag{24}$$

$$\frac{d^2}{d\mathbf{x}_j^2}\mathbf{u}(t, \mathbf{x}^i) = -\sum_k \left(\frac{\pi\omega_{k_j}}{L}\right)^2 \hat{\mathbf{u}}_k(t)\prod_{i=1}^d \exp(i\omega_{k_i}\mathbf{x}_i). \tag{25}$$

In general, linear translation-invariant differential operators may be obtained as a polynomial on the frequencies $\omega$, leading to the following representation for the Fourier multiplier $M$ used as part of the initialization:

$$\mathbf{F}_{\text{linear}}(\mathbf{u}, \nabla\mathbf{u}, \nabla\nabla\mathbf{u}, \ldots) := a_0\mathbf{u} + \sum_i a_{1i}\frac{d}{d\mathbf{x}_i}\mathbf{u} + \sum_{i,j} a_{2ij}\frac{d^2}{d\mathbf{x}_i d\mathbf{x}_j}\mathbf{u} + \cdots, \tag{26}$$

$$\hat{\mathbf{F}}_{\text{linear}}(\hat{\mathbf{u}}) = M \odot \hat{\mathbf{u}}, \text{ with } M_k = a_0 + \sum_i a_{1i}\left(\frac{i\pi\omega_{k_i}}{L}\right) - \sum_{i,j} a_{2ij}\left(\frac{\pi^2\omega_{k_i}\omega_{k_j}}{L^2}\right) + \cdots, \tag{27}$$

i.e. the k-indexed element of the multiplier $M$ is given as a polynomial over the frequencies $\omega$. This multiplier may then be used for the initialization procedure described previously. For example, the Laplacian operator in 2 space dimensions may be represented as:

$$M_k = -\left(\frac{\pi^2\omega_{k_1}^2}{L^2} + \frac{\pi^2\omega_{k_2}^2}{L^2}\right). \tag{28}$$

Analogous schemes are also used for the sine and Fourier basis.

---

[2]This can be easily seen from the Fourier convolution theorem as it appears when expressing nonlinear or non-translation-invariant equations in the Fourier basis: pointwise products become convolutions involving coefficients that are possibly far apart, resulting in dense connectivity matrices.

## B.2 Dimension-wise layers for multiple space dimensions

Spectral methods inherently induce dense interactions between coefficients, even across distant modes. However, in multi-dimensional settings, employing fully dense layers becomes computationally prohibitive. In particular, we can discuss the case of the 2D wave propagation problem presented in the experiments section: note that discretizing each dimension with just 100 spectral coefficients yields a total of $10^4$ basis elements. A single dense linear layer operating on this space would then require $O(10^8)$ parameters. As a result, when such layers are applied repeatedly (as would be done by a Neural ODE using it for a vector field), the computational cost becomes excessive.

Similar challenges in computer vision have led to the development of convolutional neural networks (CNNs), which employ finite-support kernels to create localized linear connections among neighboring nodes. However, this approach is ill-suited to our setting. First, CNNs are built to embed the translation invariance inherent to image recognition tasks, which is a property that does not generally apply to our problems. Moreover, their local receptive fields restrict long-range interactions, which are essential in spectral representations.

Instead, it is necessary to construct a layer that performs the dense and global-reaching connections associated with spectral methods while remaining parameter-light. We have opted for low-rank, *dimension-wise linear layers*, as shown in Figure 8.

These networks consist essentially of a Hadamard (element-wise) product followed by alternating dense linear transformations applied row-wise and column-wise. This structure can be efficiently implemented by combining tensor transpositions with batched matrix multiplication operations available in PyTorch and similar deep-learning frameworks (see Algorithm 1). For instance, given 2D coefficient matrices $\hat{\mathbf{u}}(t) \in \mathbb{R}^{m \times n}$ and layer parameters $A \in \mathbb{R}^{m \times n}$, $B \in \mathbb{R}^{n \times n}$, and $C \in \mathbb{R}^{m \times m}$, this yields:

**1. Element-wise scaling (Hadamard product)**:

$$\hat{\mathbf{u}} \mapsto \hat{\mathbf{u}} \odot A$$

where each element $\hat{\mathbf{u}}_{ij}$ is multiplied by the corresponding $A_{ij}$.

**2. Row-wise linear transformation**:

$$\begin{bmatrix} r_1 \\ r_2 \\ \vdots \\ r_m \end{bmatrix} \mapsto \begin{bmatrix} r_1 B \\ r_2 B \\ \vdots \\ r_m B \end{bmatrix}$$

where each row vector $r_i$ is multiplied by the matrix $B$.

**3. Column-wise linear transformation**:

$$\begin{bmatrix} c_1 & c_2 & \cdots & c_n \end{bmatrix} \mapsto \begin{bmatrix} Cc_1 & Cc_2 & \cdots & Cc_n \end{bmatrix}$$

where each column vector $c_j$ is multiplied by the matrix $C$.

The matrix $\hat{\mathbf{u}}$ can be visualized as

$$\hat{\mathbf{u}} = \begin{bmatrix} \hat{\mathbf{u}}_{1,1} & \hat{\mathbf{u}}_{1,2} & \dots & \hat{\mathbf{u}}_{1,n} \\ \hat{\mathbf{u}}_{2,1} & \hat{\mathbf{u}}_{2,2} & \dots & \hat{\mathbf{u}}_{2,n} \\ \vdots & \vdots & \ddots & \vdots \\ \hat{\mathbf{u}}_{m,1} & \hat{\mathbf{u}}_{m,2} & \dots & \hat{\mathbf{u}}_{m,n} \end{bmatrix} = \begin{bmatrix} r_1 \\ r_2 \\ \vdots \\ r_m \end{bmatrix} = \begin{bmatrix} c_1 & c_2 & \cdots & c_n \end{bmatrix}.$$

Each of these layers applied to a 2D spatial domain of dimensions $m, n$ requires $O(mn)$ parameters and operations, in contrast to the $O(m^2 n^2)$ complexity of a naive fully dense layer. This approach allows us to leverage the "densification" inherent to spectral methods, enhancing deep learning performance, while reducing computational and memory demands. In practice, we have found these layers to be effective drop-in replacements for dense linear layers when handling large, multi-dimensional inputs.

**Algorithm 1** Implementation of dimension-wise layers for a two dimensional input

$\hat{\mathbf{u}} \leftarrow \texttt{input}()$ $\qquad\qquad\qquad\qquad\qquad\qquad\qquad\qquad\qquad\qquad\qquad\qquad \triangleright\, \hat{\mathbf{u}} \in \mathbb{R}^{m \times n}$
$\hat{\mathbf{u}} \leftarrow \hat{\mathbf{u}} \odot A$ $\qquad\qquad\qquad\qquad\qquad \triangleright\, \text{Hadamard (pointwise) product, } A \in \mathbb{R}^{m \times n}$
$\hat{\mathbf{u}} \leftarrow \texttt{linear}(\hat{\mathbf{u}}, B)$ $\qquad\qquad\qquad \triangleright\, \text{Batched matrix multiplication, } B \in \mathbb{R}^{n \times n}$
$\hat{\mathbf{u}} \leftarrow \hat{\mathbf{u}}^T$ $\qquad\qquad\qquad\qquad\qquad\qquad\qquad\qquad\qquad\qquad\qquad\quad \triangleright\, \text{Transpose}$
$\hat{\mathbf{u}} \leftarrow \texttt{linear}(\hat{\mathbf{u}}, C)$ $\qquad\qquad\qquad \triangleright\, \text{Batched matrix multiplication, } C \in \mathbb{R}^{m \times m}$
$\hat{\mathbf{u}} \leftarrow \hat{\mathbf{u}}^T$ $\qquad\qquad\qquad\qquad\qquad\qquad\qquad\qquad\qquad\qquad\qquad\quad \triangleright\, \text{Transpose}$
$\hat{\mathbf{u}} \leftarrow \texttt{act}(\hat{\mathbf{u}})$ $\qquad\qquad\qquad\qquad\qquad\qquad\qquad\qquad\qquad\quad \triangleright\, \text{Nonlinear Activation}$

As illustrated by comparing Figures 7 and 8, these layers stand in contrast to convolutional layers, yet serve a complementary purpose. While convolutional layers impose sparsity through local connectivity and translation invariance, dimension-wise linear layers aim to preserve sparsity while retaining long-ranging connections. They can also be interpreted as low-rank tensor applications or Kronecker products.

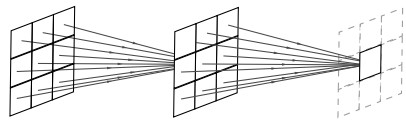

Figure 7: Schematic for convolutional linear layers.

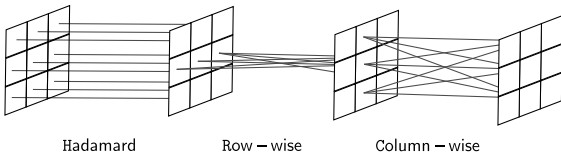

Figure 8: Schematic for dimension-wise linear transformations.

## C   Experimental details

In this section, we provide further information concerning hardwares and licensing. We also discuss additional details on the numerical experiments presented in Section 3 of the paper.

**Hardware and resources.**   All experiments were executed on identical machines, containing an Nvidia RTX 4090 GPU with 24GB VRAM, an Intel i9-13900K processor, and 128GB RAM.

**Licenses.**   The code is implemented in Python using the libraries PyTorch (version 2.1.0) and torchdyn (version 1.0.6), distributed under the BSD 3-Clause and Apache licenses, respectively.

**General setup for numerical experiments.**   All experiments are evaluated against a ground-truth solution obtained using a state-of-the-art numerical method. For each model and experiment, we report the error metrics rMAE (relative $L_1$ error) and rMSE (relative $L_2$ error) between the predicted solution $u_{\text{pred}}$ and the ground-truth solution $u_{\text{GT}}$, computed over $N$ evaluation pairs $(t_i, \mathbf{x}_i)$, as follows:

$$\text{rMAE} = \frac{\sum_{i=1}^{N} |u_{\text{pred}}(t_i, \mathbf{x}_i) - u_{\text{GT}}(t_i, \mathbf{x}_i)|}{\sum_{i=1}^{N} |u_{\text{GT}}(t_i, \mathbf{x}_i)|}, \tag{29}$$

$$\text{rMSE} = \sqrt{\frac{\sum_{i=1}^{N} (u_{\text{pred}}(t_i, \mathbf{x}_i) - u_{\text{GT}}(t_i, \mathbf{x}_i))^2}{\sum_{i=1}^{N} (u_{\text{GT}}(t_i, \mathbf{x}_i))^2}}. \tag{30}$$

In the particular case of the Burgers' equation, where the output is a 2D vector $(u, v)$, the rMAE and rMSE are computed separately for each component, and their average is reported as the final error metric.

Due to memory constraints, it is not feasible to feed the entire spatial grid to the MLP-based architectures for 2D equations. Instead, we adopted random uniform sampling at every step. For PINN, QRes, and FLS models, we sample $10,000$ points for the PDE residual, $1000$ points for the initial condition and $500$ points for the boundary condition. In contrast, PINNsFormer generates a temporal sequence of five collocation points for each sample, which are then processed through its encoder-decoder architecture. To ensure comparable memory usage and training time with the other baseline methods, we train PINNsFormer on the Burgers' and wave equations using $2,000$ points for the PDE loss, $200$ for the initial condition, and $100$ for the boundary condition. This corresponds to one-fifth of the amount sampled for other baseline methods.

### C.1 Wave equation

For the wave equation (11), by introducing $v = \dfrac{\partial u}{\partial t}$, we can rewrite the problem as a system of first-order equations (with respect to time) given by

$$
\begin{cases}
\dfrac{\partial}{\partial t} u = v \,, \\[2mm]
\dfrac{\partial}{\partial t} v = c^2(\mathbf{x}) \Delta u \,.
\end{cases}
\tag{31}
$$

The vector field to be learned by NeuSA is defined as:

$$
\hat{\mathbf{F}}_\theta(\hat{\mathbf{u}}) = \begin{pmatrix} \hat{v} \\ M \odot \hat{u} + \epsilon \mathcal{F}_\theta(\hat{u}) \end{pmatrix},
\tag{32}
$$

where $\hat{\mathbf{u}} = (\hat{u}, \hat{v})$, the weight $\epsilon = 1$ and the entries of the matrix $M$ as defined in (28). Using a simplified version of $\mathcal{F}_\theta$ that takes only $\hat{u}$ as input accelerates learning, as $\hat{v}$ does not influence the equation, and also enables a more compact model. The velocity fields may be visualized below, in Figure 9.

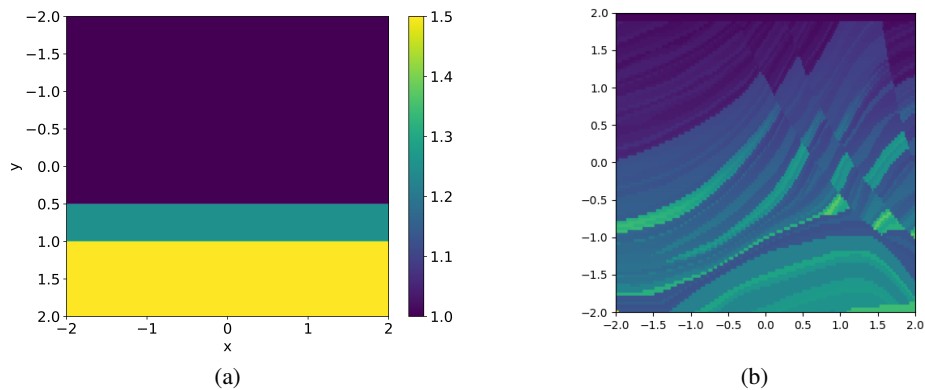

(a)                                                      (b)

Figure 9: The 2D three-layer and Marmousi mediums.

To train the NeuSA model, we simulate an infinite domain by extending the original spatial region from $[-2, 2]^d$ to $[-4, 4]^d$. The extensions of the original domain are designed to ensure that, within the simulation's temporal window, any reflected waves do not re-enter the original region of interest. This effectively prevents boundary artifacts from interfering with the solution. We use $201^d$ basis elements and integrate over 201 time steps, creating a grid with a spatial step of $\Delta x = \Delta y = 0.04$ and a temporal step of $\Delta t = 0.01$. For the spectral decomposition, we adopt a cosine basis.

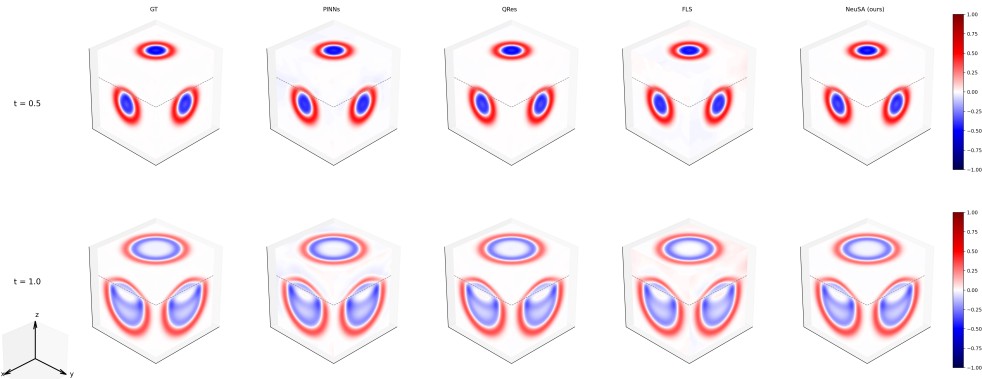

Figure 10: Solutions for the 3D wave experiment.

For the baseline models, we set the initial condition weight to $10^3$ (on both Dirichlet boundary condition and first-order initial condition, see Eq. (11) in the main text). This weighting led to consistently improved accuracy across experiments.

The ground-truth solution is calculated using a standard finite central difference scheme for the second derivatives with order 8 in the space and order 2 in time [74], with a spatial step of 0.01 and a time step of 0.001. The domain was also extended to simulate an infinite domain.

## C.2  1D sine-Gordon equation

Let us define again $v = \dfrac{\partial u}{\partial t}$ in the sine-Gordon equation (12), which allows us to rewrite it as an evolutionary system of the form

$$
\begin{cases}
\dfrac{\partial}{\partial t} u = v \,, \\[2mm]
\dfrac{\partial}{\partial t} v = \Delta u - 10 \sin(u) \,.
\end{cases}
\tag{33}
$$

From this and Eq. (12), the vector field to be learned from NeuSA can be defined as:

$$
\hat{\mathbf{F}}_\theta(\hat{\mathbf{u}}) = \begin{pmatrix} \hat{v} \\ M \odot \hat{u} + \epsilon \mathcal{F}_\theta(\hat{u}) \end{pmatrix}
\tag{34}
$$

where the weight $\epsilon$ is set to $0.1$, and the entries of the matrix $M$ are taken as defined in (28).

Owing to its architectural design, NeuSA automatically satisfies the initial condition, while the use of a sine basis ensures compliance with the boundary condition. For the baseline models, we adopted a weight of $10^3$ to the initial condition, as this choice yielded the most accurate results in our preliminary experiments.

In contrast to the experiments involving 2D equations, training the baselines for the sine-Gordon equation can be performed using the full spatial grid at each step. Since PINNsFormer produces 5 training points per grid location, it is trained on a coarser regular grid of size $101 \times 101$, whereas all other models are trained on a finer $201 \times 201$ grid. This choice of grid size presented a reasonable balance of training time, accuracy, and memory usage.

The ground-truth solution was computed through a pseudo-spectral method combined with a 4th-order Runge-Kutta integrator. We used a time step of $\Delta t = 0.001$ and a spatial resolution of $\Delta x = 0.04$.

## C.3  2D Burgers' equation

Two-dimensional Burgers' equation for the vector function $(u(t, x, y),\ v(t, x, y))$ can be written in the form of a system

$$\begin{cases} \dfrac{\partial}{\partial t} u = \nu \Delta u - u \dfrac{\partial u}{\partial x} - v \dfrac{\partial u}{\partial y}, \\[2mm] \dfrac{\partial}{\partial t} v = \nu \Delta v - u \dfrac{\partial v}{\partial x} - v \dfrac{\partial v}{\partial y}. \end{cases} \tag{35}$$

From this and (13), the vector field for NeuSA can be directly defined by

$$\hat{\mathbf{F}}_\theta(\hat{\mathbf{u}}) = \begin{pmatrix} \nu M \odot \hat{u} + \epsilon \mathcal{F}_\theta^u(\hat{u}, \hat{v}) \\ \nu M \odot \hat{v} + \epsilon \mathcal{F}_\theta^v(\hat{u}, \hat{v}) \end{pmatrix}, \tag{36}$$

where $M$ is given in (28), and the weight $\epsilon$ is set to 0.1. Also, $\mathcal{F}_\theta^u(\hat{u}, \hat{v})$ and $\mathcal{F}_\theta^v(\hat{u}, \hat{v})$ are the neural networks whose parameters we optimize during training, so the loss function is composed by the residues of each of them. The equal subscript $\theta$ on both is a slight abuse of the notation, since the parameters of these networks are not the same.

NeuSA inherently satisfies the initial conditions by construction (Theorem 1), and the use of a Fourier basis for the Burgers' equation ensures that periodic boundary conditions are also met automatically. For the baseline models, we set the initial and boundary conditions weights to $\lambda_{IC} = 10^2$ and $\lambda_{BC} = 1$, as these values produce the best performance in preliminary experiments.

The results presented in Table 1 for the Burgers' equation correspond to model training and evaluation over the time interval $[0, 1]$. We compare them and the extrapolation experiment with a ground-truth obtained through a pseudo-spectral method integrated with a 4th-order Runge-Kutta for the time interval $[0, 2]$. The temporal and spatial discretizations are set to $\Delta t = 0.001$ and $\Delta x = \Delta y = 0.02$, respectively.

## C.4  Quantitative results

The 2D experiments (Burgers' and wave equations) were run with 7 different seeds each, while the 1D sine-Gordon experiment was run with 3 different seeds. The relative $L_1$ and $L_2$ metrics and training times (in seconds) presented in the paper are an average over all random seeds. In Tables 2, 3, 4, 5, and 6, we list the mean and standard deviation of these values for the main experiments reported in the paper.

Table 2: 2D Layers: mean and standard deviation of rMAE, rMSE and runtime (TT).

| Model | rMAE | | rMSE | | TT | |
|---|---|---|---|---|---|---|
| | mean | std | mean | std | mean | std |
| PINN | $7.66 \times 10^{-1}$ | $1.27 \times 10^{-1}$ | $5.45 \times 10^{-1}$ | $8.31 \times 10^{-2}$ | 566 | 7.2 |
| QRes | $1.54 \times 10^{-1}$ | $3.45 \times 10^{-2}$ | $1.15 \times 10^{-1}$ | $3.10 \times 10^{-2}$ | 750 | 2.4 |
| FLS | $8.87 \times 10^{-1}$ | $1.63 \times 10^{-1}$ | $5.90 \times 10^{-1}$ | $9.96 \times 10^{-2}$ | 577 | 5.5 |
| PINNsFormer | 1.56 | $2.78 \times 10^{-1}$ | 1.07 | $1.41 \times 10^{-1}$ | $1,484$ | 3.4 |
| NeuSA (ours) | $1.02 \times 10^{-1}$ | $3.42 \times 10^{-2}$ | $7.49 \times 10^{-2}$ | $2.24 \times 10^{-2}$ | 530 | 1.4 |

Table 3: Marmousi: mean and standard deviation of rMAE, rMSE and runtime (TT).

| Model | rMAE | | rMSE | | TT | |
|---|---|---|---|---|---|---|
| | mean | std | mean | std | mean | std |
| PINN | 1.03 | $1.48 \times 10^{-1}$ | $6.98 \times 10^{-1}$ | $9.48 \times 10^{-2}$ | 635 | 29.7 |
| QRes | $5.57 \times 10^{-1}$ | $6.03 \times 10^{-2}$ | $4.12 \times 10^{-1}$ | $3.06 \times 10^{-2}$ | 718 | 15.8 |
| FLS | $9.99 \times 10^{-1}$ | $1.50 \times 10^{-1}$ | $6.84 \times 10^{-1}$ | $9.02 \times 10^{-2}$ | 648 | 23.9 |
| NeuSA (ours) | $2.20 \times 10^{-1}$ | $4.43 \times 10^{-2}$ | $1.71 \times 10^{-1}$ | $3.45 \times 10^{-2}$ | 573 | 2.28 |

Table 4: 3D Layers: mean and standard deviation of rMAE, rMSE and runtime (TT).

| Model | rMAE | | rMSE | | TT | |
|---|---|---|---|---|---|---|
| | mean | std | mean | std | mean | std |
| PINN | $1.86 \times 10^{-1}$ | $6.07 \times 10^{-2}$ | $7.32 \times 10^{-2}$ | $1.75 \times 10^{-2}$ | 5990 | 37.3 |
| QRes | $4.23 \times 10^{-2}$ | $7.40 \times 10^{-3}$ | $2.12 \times 10^{-2}$ | $2.60 \times 10^{-3}$ | 8775 | 30.2 |
| FLS | $1.70 \times 10^{-1}$ | $4.46 \times 10^{-2}$ | $6.95 \times 10^{-2}$ | $1.32 \times 10^{-2}$ | 6179 | 41.0 |
| NeuSA (ours) | $1.18 \times 10^{-2}$ | $2.50 \times 10^{-3}$ | $8.40 \times 10^{-3}$ | $1.80 \times 10^{-3}$ | 702 | 7.3 |

Table 5: 1D Sine–Gordon: mean and standard deviation of rMAE, rMSE and runtime (TT).

| Model | rMAE | | rMSE | | TT | |
|---|---|---|---|---|---|---|
| | mean | std | mean | std | mean | std |
| PINN | $1.70 \times 10^{-1}$ | $3.31 \times 10^{-2}$ | $1.39 \times 10^{-1}$ | $4.70 \times 10^{-3}$ | 976 | 41.4 |
| QRes | $2.58 \times 10^{-2}$ | $4.50 \times 10^{-3}$ | $1.99 \times 10^{-2}$ | $2.80 \times 10^{-3}$ | $1,315$ | 49.0 |
| FLS | $1.43 \times 10^{-1}$ | $7.20 \times 10^{-3}$ | $1.35 \times 10^{-1}$ | $2.01 \times 10^{-2}$ | $1,015$ | 68.3 |
| PINNsFormer | $7.85 \times 10^{-1}$ | $4.95 \times 10^{-1}$ | $6.81 \times 10^{-1}$ | $3.31 \times 10^{-1}$ | $3,333$ | 186.6 |
| NeuSA (ours) | $1.23 \times 10^{-3}$ | $1.49 \times 10^{-5}$ | $9.16 \times 10^{-4}$ | $5.98 \times 10^{-6}$ | 215 | 1.3 |

Table 6: 2D Burgers: mean and standard deviation of rMAE, rMSE and runtime (TT).

| Model | rMAE | | rMSE | | TT | |
|---|---|---|---|---|---|---|
| | mean | std | mean | std | mean | std |
| PINN | $1.65 \times 10^{-1}$ | $8.91 \times 10^{-2}$ | $2.21 \times 10^{-1}$ | $9.30 \times 10^{-2}$ | 871 | 2.9 |
| QRes | $2.32 \times 10^{-2}$ | $1.90 \times 10^{-3}$ | $7.27 \times 10^{-2}$ | $1.60 \times 10^{-3}$ | $1,135$ | 2.3 |
| FLS | $1.47 \times 10^{-1}$ | $7.68 \times 10^{-2}$ | $2.02 \times 10^{-1}$ | $8.08 \times 10^{-2}$ | 885 | 0.3 |
| PINNsFormer | 1.06 | $1.91 \times 10^{-1}$ | 1.05 | $1.71 \times 10^{-1}$ | $2,294$ | 108.1 |
| NeuSA (ours) | $7.66 \times 10^{-2}$ | $1.96 \times 10^{-2}$ | $1.15 \times 10^{-1}$ | $1.71 \times 10^{-2}$ | 62 | 0.2 |

# D  Additional experiments

## D.1  Number of frequencies and Training time

Due to its built-in integration process, NeuSA's convergence depends on guaranteeing the numerical stability of the Neural ODE. In particular, the more basis elements are used, generally the smaller the time-steps for the integration should be. This introduces a trade-off between spatial resolution and speed. We exemplify this behavior through the following additional experiment: We consider the 1D Burgers Equation, with initial conditions similar to those in the main paper, and evaluate how long it takes to train NeuSA with a varying number of frequencies. The results can be seen Table 7.

Table 7: Number of frequencies vs rMSE and training time. More elements are more accurate, but slower.

| Number of frequencies | rMSE | Training time (s) |
|---|---|---|
| 61 | 7.6e-2 | 109 |
| 81 | 5.6e-2 | 146 |
| 101 | 4.3e-2 | 183 |
| 121 | 3.3e-2 | 223 |
| 141 | 2.6e-2 | 264 |

## D.2 Inference time

As discussed, NeuSA's inference time is dominated by the process of integrating the Neural ODE. In essence, inference and training of Neural-Spectral Architectures is relatively costly, due to the extreme effective depth of NODEs; on the other hand, one pass generates the solution at all space and time points. Nevertheless, the much faster convergence rate in terms of number of epochs generally offsets this effect.

We compare the inference time for a subset of our experiments in Table 8, where we evaluate all models over a uniform space-time grid. As can be seen from the table, NeuSA's inference time remains more or less constant across the experiments. In contrast, the baseline architectures process every point in the spatio-temporal grid simultaneously, resulting in substantially slower inference when iterating over large amounts of points, likely as a result of memory swapping. Throughout our experiments, we attempt to mitigate this effect for the reference architectures by randomly sampling the domain in the experiments with the 2D Wave/2D Burgers equations.

Table 8: Time in seconds required by each architecture to compute the solution at the grid-like collocation points used in the experiments.

| Architecture | Wave 2D ($101 \times 101 \times 201$) | Burgers 2D ($201 \times 201 \times 101$) | Sine-Gordon ($201 \times 201$) |
|---|---|---|---|
| MLP | 0.04 | 0.28 | 0.0005 |
| QRes | 0.11 | 0.34 | 0.002 |
| FLS | 0.04 | 0.28 | 0.0006 |
| PINNsFormer | 1.29 | 2.57 | 0.02 |
| NeuSA | 0.12 | 0.11 | 0.10 |

## E Limitations

As mentioned in Section 2.1, NeuSA requires the initialization of the linear model, which may not be obvious. For Burgers' equation, for example, we have obtained the best results initializing NeuSA near a solution to the associated heat equation. As it is well known, however, some linear PDEs may turn out to be stiff when expressed in the spectral domain. This stiffness may introduce instability into the integration of the Neural ODE. For example, when we apply spectral decompositions to the Korteweg–de Vries (KdV) equation [75] the resulting system of ordinary differential equations becomes essentially stiff, especially when a wide range of frequencies is considered. This will lead to excessively large eigenvalues of the matrix used in the implementation of the associated solver, and may result in a numerically unstable solution.

This numerical issue, however, does not affect a large number of equations important for various practical applications. Consider, for instance, the canonical example of the wave equation:

$$u_{tt} - \Delta u = 0 \,. \tag{37}$$

Transforming it into the first-order hyperbolic system as shown above and applying Fourier transform we arrive at the following ODE system

$$\frac{d}{dt} \begin{bmatrix} \hat{u} \\ \hat{v} \end{bmatrix} = A \begin{bmatrix} \hat{u} \\ \hat{v} \end{bmatrix}, \quad \text{where } A = \begin{bmatrix} 0 & I \\ D & 0 \end{bmatrix} \quad \text{and} \quad D = -\mathrm{diag}(1, ..., N^2) \,. \tag{38}$$

The block matrix $A$ has eigenvalues $\lambda = \pm ik$, for $k = 1, ..., N$. According to the standard stability criterion for the Runge-Kutta methods (based on the stability polynomials and eigenvalues of the equation matrix), a time step of $h = \mathcal{O}\left(\frac{1}{N}\right)$ is required for the latter system, which is significantly larger than the time step required for the ODE system associated with the KdV equation .

This indicates that the Runge-Kutta method allows for a larger time step when applied to the wave equation compared to the KdV equation, thereby improving the practicality and efficiency of our method for a broad class of problems, including those involving wave phenomena.

