# OpenReview forum: "Neuro-Spectral Architectures for Causal Physics-Informed Networks"
_NeurIPS.cc/2025/Conference — NeurIPS 2025 poster_

### Official Review · Reviewer_aWbv · 2025-06-10

**Clarity:** 2
**Significance:** 3
**Originality:** 3
**Rating:** 4
**Confidence:** 4

**Summary:**

This paper introduces a new method in the category of physics informed neural network (PINN) inspired by neural ode and spectral methods that can handle the causality problem faced by traditional PINNs. Instead of implementing the physical loss globally with time as an input variable, this method constructs a physical loss for each discretized step in time evolution, and expresses the result as an integral of time steps. The spatial part of the solution function is treated as a linear superposition of basis functions. In this case, Fourier basis are chosen as basis functions. Networks are applied to learn time-dependent coefficients at discretized time steps. This method shows better performance on three equations in comparison with other PINN approaches, and can generalize well to future steps unseen during training.

**Questions:**

1. Are the basis functions in use fixed to be Fourier basis, or trained as networks? In lines 132-133 of main text, the paper indicates that the basis is "initialized" as Fourier basis. In lines 151-152 of main text, the paper also mentions that the basis is "trained". However, in equations 9 and 10, the dependency on trained parameters $\theta$ are not present in the basis functions $\bf{b(x)}$, indicating that those functions are not trained. In equations 25 and 26 in appendix B.1.2, the derivatives of exponential functions are calculated, also indicating that the basis functions are fixed to be Fourier. Can authors give a consistent description on basis functions?
2. The description on boundary condition fulfillment is confusing. In lines 132-133 of main text, the paper proposes to use basis functions to fulfill boundary conditions. However, boundary conditions are not satisfied with the choice of basis functions. For example, non-homogeneous Dirichlet boundary conditions should be satisfied by adding a specific term in the format of solution, so that the boundary conditions are homogenized. Can authors explain more on how boundary conditions are satisfied by the choice of basis functions?
3. In equation 11 of main text, a wave equation is shown to be solved in $\mathbb{R}^2 \times [0, 2]$, which means in spatial dimensions, the domain expands to infinity. However, figure 8 of appendix, the spatial solution domain is shown to be $[-2, 2]^2$. If the spatial domain is extended to infinity, then the boundary condition is set to be 0 at infinity. If the spatial domain is now $[-2, 2]^2$, then what is the boundary condition imposed here?

**Ethical Concerns:**

["NO or VERY MINOR ethics concerns only"]

**Final Justification:**

The rebuttal from authors answer most of my questions. Therefore I raise 1 point in the rating to recommend acceptance.

**Limitations:**

The limitations of this work are discussed in the paper.

**Paper Formatting Concerns:**

There are no major formatting issues in this paper.

**Quality:**

3

**Strengths And Weaknesses:**

Strengths:

1. The proposed method shows good extrapolation ability to unseen steps in the future, indicating that the model learns real "physics".
2. A good convergence speed is achieved by this method, in comparison with other methods mentioned in this paper.
3. The causality problem is solved with an approach like Neural ODE.

Weaknesses:

1. Some parts of this paper on boundary conditions and basis functions are confusing and inconsistent. A clarification is required. Please see question section for the most concerning points for the reviewer.
2. A Runge-Kutta method is required for integrating the steps and obtain the result, which might make the inference slow.

---

> ### Author Rebuttal · Authors · 2025-07-30
>
> We thank the reviewer for their thoughtful observations and detailed feedback on the paper. We appreciate their recognition of the method's solution to the causality problem and the method's ability to extrapolate to unseen times, as well as their highlight of our convergence speed.
>
> ## Weaknesses
>
> > 1 - **Some parts of this paper on boundary conditions and basis functions are confusing and inconsistent. A clarification is required. Please see question section for the most concerning points for the reviewer.**
>
> We address these concerns in more detail below, when responding to each question individually. In short, we recognize that the description of these elements could be improved and will deliver a more detailed explanation for the paper's final version.
>
> > 2 - **A Runge-Kutta method is required for integrating the steps and obtain the result, which might make the inference slow.**
>
> This is a reasonable concern and connects to a question by reviewer 3zAY. We discuss this briefly in the paragraph starting at line 273:
>
> "Although built on the inherently slower NODE framework, NeuSA trains significantly faster than MLP-based PINNs"
>
> In essence,  inference and training of these models is indeed slower, but the much faster convergence rate (in terms of number of epochs) generally more than compensates for this effect. For the paper's final version we will include an additional snippet expanding on this point, including the following table with the inference time (in seconds) for each architecture.
>
> We compare the time needed for each architecture to predict the solution over a spacetime grid, using the same discretizations as in the experiments. The results are as follows:
>
> | Architecture | Wave 2D (101x101x201) | Burgers 2D (201x201x101) | Sine-Gordon (201x201 |
> | -------- | ------- | ------- | ------- |
> | NeuSA | 0.12 | 0.11| 0.10 |
> | MLP | 0.04 | 0.28 | 0.0005 |
> | QRes | 0.11 | 0.34 | 0.002 |
> | FLS | 0.04 | 0.28 | 0.0006 |
> | PINNsFormer | 1.29 | 2.57 | 0.02 |
>
> As can be seen from the table, NeuSA's inference time is dominated by the numerical integration process, while the baseline architectures process every point in the spatio-temporal grid simultaneously. This results in slower inference when iterating over large amounts of points. Throughout our experiments, we mitigate this effect for the reference architectures by randomly sampling the domain in the experiments with the 2D Wave/2D Burgers equations.
>
> ## Questions
>
> > 1 - **Are the basis functions in use fixed to be Fourier basis, or trained as networks? In lines 132-133 of main text, the paper indicates that the basis is "initialized" as Fourier basis. In lines 151-152 of main text, the paper also mentions that the basis is "trained". However, in equations 9 and 10, the dependency on trained parameters  are not present in the basis functions, indicating that those functions are not trained. In equations 25 and 26 in appendix B.1.2, the derivatives of exponential functions are calculated, also indicating that the basis functions are fixed to be Fourier. Can authors give a consistent description on basis functions?**
>
> We agree with the reviewer that the discussion of classical bases and their trainable counterparts in the paper requires an improvement, and we will address this in our final revision.
>
> In our experiments, we have focused on using the Fourier bases with fixed frequencies. Though at first we attempted to use trainable bases, a more detailed ablation showed this approach underperformed when compared with a fixed basis. We came to the conclusion that these tools demanded a more thorough theoretical and practical analysis, and will add this as a future direction of research.
>
> > 2 - **The description on boundary condition fulfillment is confusing. In lines 132-133 of main text, the paper proposes to use basis functions to fulfill boundary conditions. However, boundary conditions are not satisfied with the choice of basis functions. For example, non-homogeneous Dirichlet boundary conditions should be satisfied by adding a specific term in the format of solution, so that the boundary conditions are homogenized. Can authors explain more on how boundary conditions are satisfied by the choice of basis functions?**
>
> We appreciate this remark. In our study, we restricted our attention to periodic and homogeneous Dirichlet and Neumann conditions, for which enforcing the boundary conditions through the choice of basis functions is straightforward. We will clarify that the boundary conditions are homogeneous in the final version.
>
> We also note that when the domain geometry or boundary conditions cannot be enforced automatically by the choice of basis, NeuSA may still enforce these constraints by means of Physics-Informed loss terms, as used by normal PINN architectures. We also intend to emphasize this aspect more clearly.
>
> > 3 - **In equation 11 of main text, a wave equation is shown to be solved in $\mathbb{R}^2 \times [0,2]$, which means in spatial dimensions, the domain expands to infinity. However, figure 8 of appendix, the spatial solution domain is shown to be $[-2,2]^2$. If the spatial domain is extended to infinity, then the boundary condition is set to be 0 at infinity. If the spatial domain is now $[-2,2]^2$, then what is the boundary condition imposed here?**
>
> This is a pertinent observation. The setup we are investigating here is very common in the seismics community, which is to approximate solutions to the wave equation over an infinite domain by solving it on a large domain with homogeneous Dirichlet or Neumann boundary conditions. This approach is valid as long as the domain is large enough so that the waves do not reach the boundary. Alternative approaches, including using absorbing boundary conditions, are an important direction of future research.
>
> We will clarify this in the final version of the paper and explain more carefully the relationship between the experiment's domain and the original physical problem.

---

> > ### Comment · Reviewer_aWbv · 2025-08-05
> > **Response**
> >
> > The reviewer would like to thank the authors for their detailed explanations to the concerning points. They answer most of my questions. Therefore, I would recommend acceptance for this manuscript.

---

### Official Review · Reviewer_4ytc · 2025-06-25

**Clarity:** 3
**Significance:** 3
**Originality:** 3
**Rating:** 4
**Confidence:** 4

**Summary:**

The paper introduces NeuSA (Neuro-Spectral Architectures), a new class of Physics-Informed Neural Networks (PINNs) designed to solve complex partial differential equations (PDEs), including time-dependent, nonlinear, and non-homogeneous types. Traditional MLP-based PINNs often struggle with spectral bias (favoring low frequencies) and causality violations. NeuSA overcomes these issues by:

- Projecting PDEs onto a spectral basis, capturing high-frequency dynamics.
- Integrating with Neural ODEs, which preserve temporal causality.
- Using classical spectral methods for initialization, starting training closer to the target solution.

NeuSA demonstrates superior performance on various benchmark PDEs (e.g., wave equation, Burgers’ equation, sine-Gordon equation), achieving higher accuracy, faster convergence, and better temporal consistency than other PINNs.

The architecture naturally satisfies boundary and initial conditions without extra training cost. Future work may extend NeuSA to more complex PDEs like Navier-Stokes and explore alternative spectral bases and numerical solvers to further improve performance and generalization.

**Questions:**

The authors should justify their claims by testing NeuSa in more complex setups and on a wider variety of PDEs (high dimensions, complex transient dynamics, etc.)

**Ethical Concerns:**

["NO or VERY MINOR ethics concerns only"]

**Final Justification:**

The authors' rebuttal clarified some aspects and showed further comparison with other existing models. However, most of those comparisons were limited to 1D and 2D cases. I encourage the authors to benchmark NeuSA against 3D challenging cases.

**Limitations:**

The major limitation of this work is the lack of an extended benchmark that would convince the readership of the claimed advantages of NeuSA. These kinds of benchmarks are nowadays necessary in order to position a new PINN version within the wide PINN's community.

**Quality:**

3

**Strengths And Weaknesses:**

Strengths:
the spectral formulation, alongside the introduction of NeuralODE, seem to increase the accuracy in standard benchmark and preserve causality.

Weaknesses:
NeuSA has not been tested in extreme case scenarios (very high dimensions, complex dynamics, etc.). It is therefore complex to assess NeuSA position within the vast PINN panorama.

---

> ### Author Rebuttal · Authors · 2025-07-30
>
> We thank the reviewer for their observations and feedback. It is encouraging for us that they appreciate the strength of our spectral formulation, and that they recognize the increase in accuracy in many problems, as well as the positive effects of our causal structure.
>
> ## Questions
>
> > **The authors should justify their claims by testing NeuSa in more complex setups and on a wider variety of PDEs (high dimensions, complex transient dynamics, etc.)**
>
> Although we understand the reviewer’s concern in testing NeuSA in more complex scenarios, we believe the equations used in the paper already pose very significant challenges to current PINN architectures, as pointed out by Reviewer 3zAY. Likewise, we believe the weak performance of many of the reference architectures on the chosen benchmarks, which include reflections and shocks, demonstrates that the chosen benchmarks do in fact display complex dynamics.
>
> We point out that these benchmarks go substantially beyond those used in the introduction papers for our baselines. These often deal with linear, translation-invariant problems, which NeuSA can solve at initialization to near-perfect accuracy by construction. Furthermore, most studies proposing new strategies for solving PINNs do not address high-dimensional time-dependent problems. This includes works such as QRes [1], sf-PINN [2], RoPINN [3], and PINNsFormer [4]. In particular, the PINNacle benchmark [5] notes that "when systems have higher dimensions (3D Poisson equation) or longer time spans (2D Heat equation), all methods fail to solve, highlighting the difficulties associated with complex geometries and multi-scale systems."
>
> Nevertheless, we believe the concern about the architecture's performance in high-dimensional problems is valid. To demonstrate NeuSA’s capabilities in higher dimensions, we have run an additional experiment on the wave equation in three spatial dimensions over a two-layered medium. The results can be found in the table below:
>
> | Model | Loss | rMAE | rMSE | Training Time (s) |
> |---|---|---|---|---|
> | PINN | $0.371$ | $0.145$ | $0.0613$ | $6168$ |
> | QRes | $0.078$ | $0.0298$ | $0.0167$ | $8758$ |
> | FLS | $0.313$ | $0.124$ | $0.0563$ | $6288$ |
> | NeuSA| $0.00107$ | $0.00561$ | $0.00435$ | $737$|
>
> As we can see in the table above, Neuro-Spectral Architectures are able to achieve the smallest rMSE with a much smaller training time than any other reference architecture. This experiment and a longer discussion of its results will be included in the final version of the paper.
>
> We also appreciate the reviewer’s comment regarding the need to assess NeuSA in a wider range of problems. We would like to emphasize, however, that the current benchmarks cover a relatively wide range of problems and in many ways improve upon basic test cases typically found in the PINNs literature. In particular, our numerical tests consist of:
>
> 1. The 2D wave equation in heterogeneous media, which introduces discontinuities in wave speed and demands high-frequency resolution to capture reflections and mode conversions accurately. Our benchmarks address substantially more complex geometries when compared to those reported in [2,3,4], all of which focus on the 1D wave equation in a homogeneous medium. Our benchmark is also more complex than its analogue in PINNacle [5], due to the presence of reflections.
>
> 2. The 1D sine-Gordon equation, which is characterized by strong nonlinearity and has proved to be a challenging test for the baseline architectures. This equation can be regarded as a more difficult, nonlinear counterpart to the 1D linear wave equation considered in many of our references.
>
> 3. The 2D Burgers’ equation with periodic initial condition, which captures nonlinear advection and the evolution of complex flow structures. This generalizes the 1D problem as it appears in QRes [5], and is almost identical to the setup in PINNacle [4].
>
> The examples provided in the paper were carefully selected to evaluate NeuSA's ability to overcome key limitations of standard PINNs, including spectral bias and poor performance in time-dependent problems. We have also focused on PDEs of practical interest with nonlinearities and inhomogeneous coefficients. Many of the other PDEs listed in the PINNacle benchmark [4] are linear and translation invariant; as mentioned above, NeuSA could solve them with zero training time, which we believe would be an unfair comparison.
>
> To further demonstrate NeuSA’s performance on challenging benchmarks, we have run a second additional experiment on the 2-D wave equation, considering the Marmousi model [6]. This is a highly irregular acoustic model that induces highly complex reflections. This problem is a canonical benchmark in acoustic geophysics and is challenging for even specialized PINNs [7].
>
> We tested NeuSA on the 2D wave equation using a cropped version of the Marmousi model. By doubling the
> frequency content and training for 3000 epochs, we achieved an RMSE of 9.05e-2 with a training time of 923 seconds. This result is comparable to the framework tackled by [8], whose architecture is specialized to simulating seismic wavefields for nonsmooth complex media. This model has a much larger set of highly optimized hyperparameters and was trained using time-domain decomposition over 45000 epochs with 80000 collocation points, and requiring hours of training in our current setup. We therefore conclude that NeuSA has a strong performance in this scenario when compared to most of the literature.  We will include these results to the final version of our paper.
>
> ### References
>
> [1] Bu, J., & Karpatne, A. (2021). **Quadratic residual networks: A new class of neural networks for solving forward and inverse problems in physics involving pdes.** In Proceedings of the 2021 SIAM International Conference on Data Mining (SDM) (pp. 675-683). Society for Industrial and Applied Mathematics.
>
> [2] Wong, J. C., Ooi, C. C., Gupta, A., & Ong, Y. -S. (2024) **Learning in Sinusoidal Spaces With Physics-Informed Neural Networks.** IEEE Transactions on Artificial Intelligence, vol. 5, no. 3, pp. 985-1000.
>
> [3] Wu, H., Luo, H., Ma, Y., Wang, J., & Long, M. (2024). **ROPINN: Region optimized physics-informed neural networks.** Advances in Neural Information Processing Systems, 37, 110494-110532.
>
> [4] Zhao, Z., Ding, X., & Prakash, B. A. (2023). **PINNsformer: A transformer-based framework for physics-informed neural networks.** arXiv preprint arXiv:2307.11833.
>
> [5] Zhongkai, H., Yao, J., Su, C., Su, H., Wang, Z., Lu, F., Xia, Z., Zhang, Y., Liu, S., Lu, L., & Zhu, J. (2024). **PINNacle: A comprehensive benchmark of physics-informed neural networks for solving pdes.** Advances in Neural Information Processing Systems, 37, 76721-76774.
>
> [6] Brougois, A., Bourget, M., Lailly, P., Poulet, M., Ricarte, P., & Versteeg, R. (1990). **Marmousi, model and data.** In EAEG workshop-practical aspects of seismic data inversion (pp. cp-108). European Association of Geoscientists & Engineers.
>
> [7] Mishra, S., & Shekar, B. (2025). **Wave Equation and Neural Solvers: What Works and What Needs to Change?.** In 86th EAGE Annual Conference & Exhibition (Vol. 2025, No. 1, pp. 1-5). European Association of Geoscientists & Engineers.
>
> [8] Ding, Y., Chen, S., Miyake, H., & Li, X. (2025). **Physics-informed Neural Networks with Fourier Features for Seismic Wavefield Simulation in Time-Domain Nonsmooth Complex Media.** IEEE Transactions on Geoscience and Remote Sensing.

---

> > ### Comment · Reviewer_4ytc · 2025-08-05
> > **Further comments**
> >
> > I'd love to thank the reviewers for their insightful rebuttal. The NeuSA performances are quite impressive compared to the SOTA.
> > Out of curiosity, what are the metrics for Marmousi, in terms of rMAE, rMSE, rRMSE? In their 2025 paper (https://doi.org/10.1016/j.jcp.2025.113813), Lehmann et al. trained an F-FNO for 3D elastodynamics wave propagation and tested it on the Marmousi model. After fine tuning on slight random variations of the initial Marmousi model, also adopting random sources, up to 1000 new samples. the authors obtained and rMAE 0.16 and rRMSE 0.24. I think this can be used as benchmark for NeuSA so to test it on extreme case scenarios.

---

> ### Author Response · Authors · 2025-08-06
> **Metrics for Marmousi**
>
> We thank the reviewer for their feedback and their comparison between NeuSA and the SOTA. The metrics obtained for Marmousi [6] were an rMAE of 0.139 and an rMSE of 0.0905. We note that we follow the evaluation protocol of RoPINN [3] and PINNacle [5], with the equivalence between rMAE and rL1RE, and rMSE and rL2RE, as shown in Appendix C, equations (30) and (31). There is an alternate definition presented in PINNsFormer [4] that defines the rRMSE as what we previously defined as rMSE, which has:
>
> $rRMSE = \sqrt{\frac{\sum_{i = 1}^N (u_{pred}(\textbf{t},\textbf{x}) - u_{GT}(\textbf{t},\textbf{x}))^2}{\sum_{i = 1}^N (u_{GT}(\textbf{t},\textbf{x}))^2}}$
>
> This gives an alternate definition for the rMSE given by:
>
> $rMSE = \frac{\sum_{i = 1}^N (u_{pred}(\textbf{t},\textbf{x}) - u_{GT}(\textbf{t},\textbf{x}))^2}{\sum_{i = 1}^N (u_{GT}(\textbf{t},\textbf{x}))^2} = rRMSE^2$
>
> When considering these alternate definitions, we have an rMSE of 0.00819 and an rRMSE of 0.0905. We note that this is not the same context as in Lehman et al.: They consider a 3D version of Marmousi [6] and use the neural operator framework, trained with simulation data; we consider for this experiment a 2D version trained purely through physics-informed means. The metrics that Lehman et al. use are also not the same as the ones we used, as they have yet another definition of rMAE and rRMSE. Nevertheless, we agree that this context could be an interesting scenario for future testing, and thank the reviewer for bringing this reference to our attention.

---

> > ### Comment · Reviewer_4ytc · 2025-08-08
> >
> > Thanks for addressing my comments. I've adjusted my ratings accordingly.

---

### Official Review · Reviewer_gAMj · 2025-06-30

**Clarity:** 4
**Significance:** 3
**Originality:** 4
**Rating:** 5
**Confidence:** 3

**Summary:**

The paper introduces NeuSA (Neurospectral Architectures) as a novel class of PINN architecture. This class satisfies by construction the boundary/initial conditions of the PDE, which alleviates one of the main training difficulty of PINNs, given by the imbalance in the loss components. Moreover, time dependency is resolved through continuous time integration based on a NeurODE formulation. The model also satisfies causality and uniqueness of the solution, which are two very interesting properties for a Physics-Informed model. The proposed NeuSA is validated on some benchmark PDEs and compared with state-of-the-art architectures.

**Questions:**

* I find it surprising that QRes outperforms NeuSA in the relative L2 Error over time (except far from the extrapolation region) in Figure 5, do you have some intuition on why this happens?
* I was wondering whether the authors tested also other training protocols (as quasi newton methods) which might enhance the performance of the Model in the training region.
* Related to the question above, I would be curious to know (if the authors have tested it) what would be the extrapolation performance of the model when a NeuSA is trained to very good accuracy (say, around 10^-6 error) in the training region. Would the extrapolation error increase or stay limited?

**Ethical Concerns:**

["NO or VERY MINOR ethics concerns only"]

**Final Justification:**

The discussion with the authors confirmed my score.

**Limitations:**

The authors have addressed the limitations throughout the paper.

**Paper Formatting Concerns:**

I have no concerns on the paper formatting,

**Quality:**

3

**Strengths And Weaknesses:**

The paper is in a very good format. In particular:
* The model is well presented, and the architectural choices are well motivated. Overall, the paper is well written.
* I find the combination of the PINN framework with the concept of NeurODE for the time integration very elegant.
* The model tackles by construction several long-standing issues of the PINN architecture.
* The achieved performance in training time is surprising, considering the time integration chosen.

On the other hand, there are some minor issues that concerns me:
* The PDEs considered for testing seem very suitable for the chosen architecture, due to the choice of a Fourier basis.
* The model only considers simple geometries, while I think the real challenge in imposing boundary conditions typically lies in more complex ones

---

> ### Author Rebuttal · Authors · 2025-07-30
>
> We thank the reviewer for their insightful observations and endorsement. We appreciated that they find our architecture to be well motivated and interesting, and noted the substantially faster convergence speed of our method. We are also glad that they acknowledged the architecture's unique features, including uniqueness of trajectories and causal structure. We address in the weaknesses and questions pointed by the reviewer below.
>
> ## Weaknesses
>
> > 1. **The PDEs considered for testing seem very suitable for the chosen architecture, due to the choice of a Fourier basis.**
>
> Though we understand the reviewer's concern, we would argue that the PDEs chosen for the experiments are not particularly tied to the strengths of the Fourier basis. In particular, both the wave equation in inhomogeneous media and Burgers' equation involve highly localized phenomena, shocks and high-frequency components, which Fourier series struggle to represent.
> To illustrate this point, we will add to the final version an additional experiment on the 2-D wave equation, considering the Marmousi model [1]. This is a highly irregular acoustic model that induces highly complex reflections. This problem is a canonical benchmark in acoustic geophysics and is extremely challenging for even specialized PINNs [2].
>
> We tested NeuSA on the 2D wave equation using a cropped version of the Marmousi model. By doubling the frequency content and training for 3000 epochs, we achieved an RMSE of 9.05e-2 with a training time of 923 seconds. This result is comparable to the framework tackled by [2], whose architecture is specialized to simulating seismic wavefields for nonsmooth complex media. This model has a much larger set of highly optimized hyperparameters and was trained using time-domain decomposition over 45000 epochs with 80000 collocation points, and requiring hours of training current setup. We therefore conclude that NeuSA has a strong performance in this scenario when compared to most of the literature. We will add these results to the final version of our paper.
>
> > 2. **The model only considers simple geometries, while I think the real challenge in imposing boundary conditions typically lies in more complex ones.**
>
> We thank the reviewer for their observation. We quickly mention this in the paragraph starting on line 283:
>
> *"all experiments take place on simple spatial domains [...] more complex geometries would require the implementation of more generic bases,"*
>
> In other words, NeuSA deals most easily with simple domain shapes, whereby we may leverage the basis's properties. This may be extended in a straightforward manner by using differentiable implementations of other bases (such as cylindrical and spherical harmonics).
> On the other hand, NeuSA remains a meshless PINN, so it is still possible to enforce different boundary conditions and domain constraints through carefully crafted losses.
> We also stress that many developments in PINNs, including the benchmark paper PINNacle [4] and the reference papers for PINNs, FLS, QRes, and FLS, deal almost exclusively with domains with simple boundary shapes.
>
> ## Questions
>
> > 1. **I find it surprising that QRes outperforms NeuSA in the relative L2 Error over time (except far from the extrapolation region) in Figure 5, do you have some intuition on why this happens?**
>
> Our intuition is that this is due to the solution of Burger's equation developing shocks over time, which Fourier basis struggle to represent well. Due to its reliance on spectral bases, NeuSA cannot truly "focus" on any given part of the domain, unlike other architectures. In other words, the Fourier basis will tend to induce global components instead of local components of the solution. We have verified that the vast majority of the error is concentrated in the close vicinity of the shocks; NeuSA seems to approximate the solution quite well outside of this interval, as can be seen from the experiment results.
> On the other hand, we note that QRes is a highly nonlinear and flexible network, and is able to better represent solutions with shocks.
>
> > 2. **I was wondering whether the authors tested also other training protocols (as quasi newton methods) which might enhance the performance of the Model in the training region.**
>
> To address this question, we have run new experiments where we use the LBFGS optimizer [5], instead of ADAM, and compared training time and rMSE. The results are in the table below
>
> | Equation | LBFGS rMSE  | ADAM rMSE | LBFGS training time (s) | ADAM training time (s) |
> | -------- | ------- |  ------- |  ------- |  ------- |
> | Burgers 2d | 0.142 | 0.11 | 576 | 62 |
> | Wave 2d | 0.0502 | 0.07 | 625 | 530 |
> | Sine-Gordon | 0.00105 | 0.0009 | 831 | 215 |
>
>  As it can be seen in the table training with LBFGS often led to much slower training in exchange to little to no improvements in accuracy. We will add this table and discussion to our appendices in the final version.
> > 3. **Related to the question above, I would be curious to know (if the authors have tested it) what would be the extrapolation performance of the model when a NeuSA is trained to very good accuracy (say, around $10^-6$ error) in the training region. Would the extrapolation error increase or stay limited?**
>
> To address this question, we will add an additional experiment focusing on the performance of NeuSA for extrapolation. We trained the Sine-Gordon equation over the time interval $[0, 1.5]$ using 100 and 1000 time steps, and obtained PDE losses in this time interval of 6e-4 and 8e-6, respectively. The RMSE over the interval $[0, t_f]$ is shown in the table below:
>
> | Time Steps | Loss | $t_f = 1.5$ | $t_f = 1.6$ | $t_f = 1.7$ | $t_f = 1.8$ | $t_f = 1.9$ | $t_f = 2.0$ |
> | -------- | ------- | ------- |  ------- |  ------- |  ------- |  ------- |  ------- |
> | 100  | $6\times 10^{-4}$ | $1.04\times 10^{-3}$ | $2.14\times 10^{-3}$ |  $1.09\times 10^{-2}$ | $3.41\times 10^{-2}$ | $6.99\times 10^{-2}$ | $1.09\times 10^{-1}$ |
> | 1000 | $8\times 10^{-6}$ | $2.69\times 10^{-4}$ | $1.74\times 10^{-3}$ | $1.05\times 10^{-2}$ | $3.32\times 10^{-2}$ | $6.83\times 10^{-2}$ | $1.06\times 10^{-1}$ |
>
> These results shows that the extrapolation capacity of Neusa depends on the accuracy of the field in the training region. However, overfitting in the training region does not necessarily reduce the error in the more distant regions.
>
> ### References
>
> [1] Brougois, A., Bourget, M., Lailly, P., Poulet, M., Ricarte, P., & Versteeg, R. (1990, May). **Marmousi, model and data**. In *EAEG workshop-practical aspects of seismic data inversion* (pp. cp-108). European Association of Geoscientists & Engineers.
>
> [2] Mishra, S., & Shekar, B. (2025). **Wave Equation and Neural Solvers: What Works and What Needs to Change?**. In 86th *EAGE Annual Conference & Exhibition* (Vol. 2025, No. 1, pp. 1-5). European Association of Geoscientists & Engineers.
>
> [3] Ding, Y., Chen, S., Miyake, H., & Li, X. (2025). **Physics-informed Neural Networks with Fourier Features for Seismic Wavefield Simulation in Time-Domain Nonsmooth Complex Media**. IEEE Transactions on Geoscience and Remote Sensing.
>
> [4] Zhongkai, Hao, et al. **"Pinnacle: A comprehensive benchmark of physics-informed neural networks for solving PDEs."** Advances in Neural Information Processing Systems* 37 (2024): 76721-76774.
>
> [5] Liu, Dong C., and Jorge Nocedal. **"On the limited memory BFGS method for large scale optimization."** *Mathematical programming* 45.1 (1989): 503-528.

---

> > ### Comment · Reviewer_gAMj · 2025-08-05
> >
> > I would like to thank the authors for the effort made to address my concerns and curiosities, which have been fully covered.

---

### Official Review · Reviewer_3zAY · 2025-07-02

**Clarity:** 4
**Significance:** 4
**Originality:** 4
**Rating:** 5
**Confidence:** 5

**Summary:**

This study introduces NeuSA (Neuro-Spectral Architectures), which is an adaptation of the Neural Ordinary Differential Equation (Neural ODE) architecture to the Physics Informed Neural Network (PINN) problem - that is, approximating the solution of a given PDE by using the PDE residual in the loss function. Inspired by the spectral methods for PDEs, NeuSA models the solution as a linear combination of spatial basis functions and uses a Neural ODE to learn the time dependence of the basis coefficients. Furthermore, while the basis functions are initialized to be the Fourier basis, these are also learned during training by the use of learnable orthogonal transforms. As the basis function expansion can be set up to satisfy both the initial and boundary conditions, the loss function only contains the PDE residual term. Experiments on challenging low dimensional systems show the strong performance of NeuSA compared to the baselines considered.

**Questions:**

The first part of my questions are related to the points raised in the Weaknesses section above.

1. How does the algorithm performance and the computational overhead change with the number of basis elements? Is this a hyperparameter to be sweeped over, or do the authors have a strategy / heuristic to determine the appropriate number?

2. Related to the previous question, the initial conditions for the PDE problems in the experiments are either well described with finite Fourier basis elements (2D Burgers': product of sines/cosines; 2D wave, 1D sine-Gordon: Gaussian, whose spectrum decays as $\exp(-k^2)$ with increasing wavenumber $k$.). When the initial condition becomes more difficult to describe with a truncated Fourier series (such as a square wave), does this method still retain its performance?

3. I am curious about the per-epoch training time for each of the models considered in the paper. Is my reasoning correct, or have I made a mistake?

Some more minor questions are as follows:

4. Neural ODEs tend to struggle especially for long time series data, as the gradient pathologies become increasingly likely to occur (due to all the recurrent computation that stacks up while propagating the ODE solver forward in time) and the cost of the algorithm also scales linearly with time. In the context of NeuSA, this situation would correspond to solving a PDE over a longer temporal domain. Is this observed in practice?

5. The algorithm's initialization scheme relies on linearizing the PDE and doing some algebra to derive the Fourier multipliers. Was this done manually or automatized, for example, by symbolic calculations?

**Ethical Concerns:**

["NO or VERY MINOR ethics concerns only"]

**Final Justification:**

This is a solid paper on developing a Neural ODE-based PINN architecture that shows strong performance on challenging PDE problems. The concerns I had prior to the discussion phase were minor, and even these were well-addressed by the authors with multiple extra calculations performed. As such, I will keep my original positive score.

**Limitations:**

Yes

**Paper Formatting Concerns:**

There are no paper formatting concerns

**Quality:**

3

**Strengths And Weaknesses:**

**Strengths**

1. Strong performance on challenging problems

The problems considered in the paper such as wave propagation in heterogeneous media or 2D Burger's equation with shock development are all difficult problems that conventional PINNs tend to struggle on. However, the proposed approach returns accurate solutions for these problems with much less computational cost than its competitors.

2. Stable training dynamics due to improved architecture design

Due to the model formalism, NeuSA respects causality and does require multiple competing terms in its loss functions. This leads to faster and more stable training, as evidenced in the right panel of Figure 1.


3. Clear architectural motivations

The use of a Neural ODE-like architecture for the PINN problem is clearly motivated from the viewpoint of spectral methods for PDEs.


**Weaknesses**

1. Choosing the number of basis function elements

The number of Fourier basis elements used for training seems to be a key hyperparameter for this method, as it affects multiple aspects of training, including: 1) the maximum spatial frequency that the model output can have, 2) the computational cost of the model ($n$ basis elements for each of the $d$ spatial dimensions will scale roughly as O($n^d$), even with the dimensionwise layers used in this study), 3) how well the initial condition is described by the model. As such, an experiment to gauge the model performance as a function of number of basis elements used would provide more insights about the proposed method.

2. Confusion about the speed of the algorithm

The authors claim that despite the slowness of the Neural ODE architecture, NeuSA can achieve comparable or faster training times than competing non-Neural ODE approaches, with one of the reasons being NeuSA being able to compute the loss with a single forward pass whereas conventional approaches requiring multiple passes. While the shorter training times reported in Table 1 and Table 2,3,4 (in the Supplementary) seem to support this claim, it needs to be noted that the total training steps for NeuSA were 10 (2D Wave, 1D Sine-Gordon) or 100 times lower (2D-Burgers) than its counterparts. While impressive, this does mean that the per-step training time of NeuSA is often much higher than the other approaches (for 2D Wave, x3.57-x9.40 times slower), and the above claim does not seem to hold.

Minor

3. While recently proposed, a method for solving PDE that draws from the spectral method does exist in the literature [1]. While that approach and the current paper differs quite a bit, it would be great if the authors could incorporate the reference in their discussions, especially as the reference considers non-Fourier basis functions as well.

- [1]: J. Choi et al. Spectral operator learning for parametric PDEs without data reliance. Comput. Methods Appl. Mech. Engrg. 420, 15 (2024).

---

> ### Author Rebuttal · Authors · 2025-07-30
>
> We thank the reviewer for their endorsement and detailed analysis. We are encouraged that they find our architecture to be well motivated and interesting. We appreciate that the reviewer highlights the model's strong experimental performance and notices its substantially faster convergence speed. We respond in detail to the weaknesses and questions raised by the reviewer below. Weaknesses 1 and 2 are addressed in questions.
>
> ## Weaknesses
>
> > 3. **While recently proposed, a method for solving PDE that draws from the spectral method does exist in the literature. While that approach and the current paper differ quite a bit, it would be great if the authors could incorporate the reference in their discussions [...]**
>
> We appreciate the additional reference and will include it in the final paper. As mentioned, the approaches are quite different: The authors of SCLON are interested in operator learning for solving parametric partial differential equations (PDEs), while we address a fixed initial value problem, more in line with PINNs literature.
>
> Although they also leverage expansions in orthogonal functions, the way that time propagation is dealt with is also quite different: SCLON divides the time domain into multiple discrete subintervals and trains a separate network for each; our method, in turn, extensively uses continuous-time Neural ODEs. Nevertheless, some aspects — such as Legendre spectral methods and the sequential approach for time-dependent PDEs — could be incorporated into future extensions of the NeuSA framework.
>
> ## Questions
>
> > 1. **How does the algorithm performance and the computational overhead change with the number of basis elements? Is this a hyperparameter to be sweeped over, or do the authors have a strategy / heuristic to determine the appropriate number?**
>
> Our initial criterion was to use as many basis elements as possible for a given spatial sampling of the domain. To promote a wider discussion on the role of basis functions, we have included a small section in the supplementary material with experiments exploring this question further.
>
> We consider the 1D Burgers equation with a variable number of basis elements. To ensure numerical stability, we increase the number of time steps proportionally to the number of basis elements.  The table below shows how the relative root mean-square error and the training time are affected by the number of frequencies used.
>
> | Number of frequencies | rMSE | Training time (s) |
> | -------- | ------- | ------- |
> | 21 | 0.181 | 35 |
> | 41 | 0.109 | 72 |
> | 61 | 0.076 | 109 |
> | 81 | 0.056 | 146 |
> | 101 | 0.043 | 183 |
> | 121 | 0.033 | 223 |
> | 141 | 0.026 | 264 |
> | 161 | 0.021 | 299 |
>
> As can be seen, the number of frequencies has a substantial influence over both the error and training time, but the accuracy benefits have diminishing returns. As a general heuristic for classical spectral methods, we would expect small improvements after the basis elements chosen cover the frequency band induced by the solution; it is still unclear if or how this intuition carries over to the neural case. We will include this experiment in an appendix.
>
> > 2. **Related to the previous question, the initial conditions for the PDE problems in the experiments are either well described with finite Fourier basis elements [...] When the initial condition becomes more difficult to describe with a truncated Fourier series (such as a square wave), does this method still retain its performance?**
>
> The effects of high-frequency components on performance may be seen in the results of Experiment number 3, with the 2D Burgers' equation. The solution eventually converges to a sharp, discontinuous shock profile with high frequency content, which is widely known to be a difficult scenario for spectral and pseudo-spectral methods due to the non-localized behavior of the Fourier basis elements. Nevertheless, NeuSA ranks second among the listed architectures, while vastly outperforming the others in terms of speed.
>
> In order to investigate NeuSA's performance on initial conditions less easily described by the Fourier basis, we have performed an additional experiment on the Sine-Gordon equation, with a triangular initial condition, the quantitative results of which are shown in the table below:
>
> | Model | rMSE | Training time (s) |
> | -------- | ------- | ------- |
> | PINN | 4.47E-02 | 420 |
> | QRes | 1.27E-02 | 498 |
> | FLS | 4.43E-02 | 433 |
> | NeuSA | 2.02E-02 | 292 |
>
> We observe that NeuSA has a slightly worse performance than QRes, albeit with a smaller training time. This can be attributed to the slower decay of the triangle wave's Fourier coefficients.
>
>
> > 3. **I am curious about the per-epoch training time for each of the models considered in the paper. Is my reasoning correct, or have I made a mistake?**
>
> We discuss this briefly in the paragraph starting at line 273:
>
> *"Although built on the inherently slower NODE framework, NeuSA trains significantly faster than MLP-based PINNs"*
>
> In essence, inference and training of Neural-Spectral Architectures is indeed slower, due to the extreme effective depth of NODEs.  On the other hand, the much faster convergence rate in terms of number of epochs generally offsets this effect.
>
> For the paper's final version we will include an additional snippet expanding on this point, as well as a table with the **average inference time (in seconds)** for each model. For the table, we include the time in seconds needed for each model to calculate the solution at the grid-like collocation points used in the experiments:
>
> | Architecture | Wave 2D (101x101x201) | Burgers 2D (201x201x101) | Sine-Gordon (201x201) |
> | -------- | ------- | ------- | ------- |
> | NeuSA | 0.12 | 0.11| 0.10 |
> | MLP | 0.04 | 0.28 | 0.0005 |
> | QRes | 0.11 | 0.34 | 0.002 |
> | FLS | 0.04 | 0.28 | 0.0006 |
> | PINNsFormer | 1.29 | 2.57 | 0.02 |
>
> As can be seen from the table, NeuSA's inference time remains more or less constant across the experiments, as it is dominated by the numerical integration process. In contrast, the baseline architectures process every point in the spatio-temporal grid simultaneously, resulting in slower inference when iterating over large amounts of points. Throughout our experiments, we mitigate this effect for the reference architectures by randomly sampling the domain in the experiments with the 2D Wave/2D Burgers equations.
>
> > 4. **Neural ODEs tend to struggle especially for long time series data, as the gradient pathologies become increasingly likely to occur (due to all the recurrent computation that stacks up while propagating the ODE solver forward in time) and the cost of the algorithm also scales linearly with time. In the context of NeuSA, this situation would correspond to solving a PDE over a longer temporal domain. Is this observed in practice?**
>
> Yes, we observed this in some experiments with very long time intervals. There are several ways to address this issue, but one approach that has been successful in our trials is to train different models over different time intervals. This increases the number of parameters and training time, but makes the model more adaptable to the different behaviors of the solution over different time intervals. Still, conventional architectures would generally fail under the same conditions, the longer time horizons inducing failure modes due to their lack of causality.
>
> > 5. **The algorithm's initialization scheme relies on linearizing the PDE and doing some algebra to derive the Fourier multipliers. Was this done manually or automatized, for example, by symbolic calculations?**
>
> This was done manually. We will state this more clearly in the final version.

---

> > ### Comment · Reviewer_3zAY · 2025-08-04
> >
> > I find that the authors have carefully addressed all of my questions.
> > I thank the authors for their efforts in performing the extra analyses.

---

### Note · Authors · 2025-08-14

We would like to thank the reviewers for their comments and suggestions. We believe their questions and insights led us to be more clear and complete, for which we are very grateful. Neuro-Spectral Architectures (NeuSA) is a novel class of physics informed neural networks that is inspired by spectral methods to solve linear and non-linear partial differential equations. These networks satisfy properties such as causality and spectral fidelity by construction. As mentioned by the reviewers, when evaluating NeuSA in a range of problems, from two-dimensional and three-dimensional wave propagation in non-homogeneous media to the two-dimensional Burger's equation, the results obtained are strong compared to the state-of-the-art, often exhibiting superior accuracy, higher temporal consistency, and faster predictive convergence.

---

### Decision · Program_Chairs · 2025-09-17

**Decision:**

Accept (poster)

**Comment:**

This paper demonstrates several important strengths that justify its acceptance. Most notably, the proposed method exhibits strong extrapolation ability to unseen future steps, indicating that it successfully captures the underlying physical principles rather than merely fitting the training data. This is a critical feature for ensuring robustness and generalizability in scientific and engineering applications. In addition, the method achieves a faster convergence speed than other approaches discussed in the paper, highlighting its efficiency and practical utility, especially for large-scale problems where computational cost is significant. Furthermore, the authors effectively resolve the causality issue by employing a strategy inspired by Neural ODEs, thereby maintaining temporal consistency and physical plausibility in the learned dynamics. Equally important, the authors have carefully and appropriately addressed the reviewers’ questions, providing clarifications and revisions that further strengthen the manuscript. Taken together, these merits make the paper a valuable contribution, and I recommend acceptance.